# Comprehensive Physiology, Cytology, and Transcriptomics Studies Reveal the Regulatory Mechanisms Behind the High Calyx Abscission Rate in the Bud Variety of Korla Pear (*Pyrus sinkiangensis* ‘Xinnonglinxiang’)

**DOI:** 10.3390/plants13243504

**Published:** 2024-12-15

**Authors:** Xian’an Yang, Shiwei Wang, Zhenbin Jiang, Cuifang Zhang, Long Zhao, Yutong Cui

**Affiliations:** 1College of Forestry and Landscape Architecture, Xinjiang Agricultural University, Urumqi 830052, China; 18095950962@163.com (X.Y.); wsw850204@163.com (S.W.); 13161097837@163.com (L.Z.); 18040796669@163.com (Y.C.); 2Key Laboratory of Forestry Ecology and Industry Technology in Arid Region, Xinjiang Agricultural University, Urumqi 830052, China; 3Forest Fruit Technology Research and Promotion Center, Bayingoleng Mongolian Autonomous Prefecture, Korla 841000, China; 18291522086@163.com

**Keywords:** Korla fragrant pear, bud mutation, calyx tube shedding, transcriptome, differentially expressed genes, weighted gene co-expression network analysis

## Abstract

Whether the calyx tube of the Korla fragrant pear falls off seriously affects the fruit quality. ‘Xinnonglinxiang’ is a mutant variety of the Korla fragrant pear, which has a high calyx removal rate under natural conditions, and calyx tube fall seriously affects the fruit quality. The mechanism behind the high calyx removal rate of ‘Xinnonglinxiang’ remains unclear; thus, Korla fragrant pear (PT) and ‘Xinnonglinxiang’ (YB) with different degrees of calyx abscission were used as examples and the abscission areas of calyx tubes were collected in the early (21 April), middle (23 April), and late (25 April) shedding stages to explore the regulatory mechanism behind the abscission. The combination of the results of physiological, cytological, and transcriptomic methods indicated the highest number of differentially expressed genes (DEGs) in the middle of shedding. GO (Gene Ontology) enrichment analysis showed that the expression levels of genes related to the CEL (cellulase) and PG (polygalacturonase) activity functional pathways differed significantly in the two varieties during the three periods, whereas Kyoto Encyclopedia of Genes and Genomes (KEGG) enrichment analysis showed that the DEGs were significantly concentrated in the plant hormone signal transduction pathway in all three periods. The expression levels of genes related to the plant hormone signal transduction pathway differed significantly for the two varieties during calyx shedding. Five gene modules were obtained using Weighted Gene Co-Expression Network Analysis (WGCNA), and transcriptome data were correlated with five physiological index values. Two key modules that highly correlated with the Eth (ethylene) response were then screened, and 20 core genes were identified, with *IRX10*, *IRX9*, and *OXI1* likely the hub genes that are involved in the regulation of calyx shedding in the YB variety. The obtained results provide reliable data for the screening of candidate genes for calyx shedding and analysis of the regulatory mechanism behind a high calyx shedding rate, providing a theoretical basis upon which the calyx shedding rate of fruits can be improved through genetic improvement.

## 1. Introduction

Korla fragrant pear (*Pyrus sinkiangensis* Yu), from the Pyrus family of Rosaceae, is a native pear variety in Korla, Xinjiang, China [1] that is exported to many countries worldwide due to its excellent fruit quality and has gradually become one of the main agricultural products exported from the region [2]. However, the low fruit calyx removal rate under natural conditions and the resulting persistence of the calyx tubes affect the shape and quality of the fruit, hindering the high-quality development of the Korla fragrant pear industry [3]. In the most persistent fruits, the calyx does not fall at all, instead developing into part of the fruit top as the young fruit grows and resulting in deformed fruit with a high stone cell content, rough texture, poor flavor, and long maturity period. This phenomenon minimizes the economic benefits of producing this type of fruit [4], as the persistence of the calyx tube not only affects the fruit quality but also affects its commodity value and market competitiveness. Based on these facts and preliminary investigation, we bred a mutant bud variety of the Korla fragrant pear (*Pyrus sinkiangensis* ‘Xinnonglinxiang’), which has the characteristic of a high calyx removal rate under natural conditions and retains the original excellent quality of the Korla fragrant pear fruit. Clarifying the regulatory mechanism behind the high calyx removal rate in ‘Xinnonglinxiang’ is of great significance and is expected to adjust the variety structure of the Korla fragrant pear, promoting the economic development of fragrant pear planting.

The calyx tube is a metamorphic organ in leaves, with the shedding of fruit tree organs a common physiological phenomenon in the plant kingdom [5] that includes the shedding of flowers, fruits, and leaves, among others. Shedding can eliminate weak organs during the normal growth and development of fruit trees and is conducive to the timely supply of nutrients, mineral elements, and plant hormones to other tissues and organs, ensuring normal growth and development [6,7]. The abscission of plant organs is a complex and systematic physiological phenomenon that originates from active changes in the abscission layer cells, which are located within an abscission zone lying between the plant body and the isolated organ, separated by an obvious boundary [8,9]. The abscission zone is usually composed of 5–50 layers comprising cells of small volume, a dense cytoplasm, and small intercellular spaces, which are denoted abscission layer cells [10,11]. When plants are affected by their own physiological regulation or stress, an abscission signal is triggered, and the balance between hormones near the abscission zone is disrupted, resulting in abscission of the organ [12,13]. Hormones such as cytokinin (CTK), abscisic acid (ABA), and ethylene (Eth) are currently thought to promote organ shedding, while others such as auxin (IAA) and gibberellin (GA_3_) are thought to inhibit organ shedding [14,15], with results indicating such associations reported in studies investigating the shedding of grapes (*Vitis vinifera*) [16], *Populus trichocarpa* leaves [17], strawberry (*Fragaria × ananassa*) fruit [18], *Ginkgo biloba* leaves [19], *Lonicera cearulea* fruit [20], *Citrus unshiu* fruit [21], and tomato (*Solanum lycopersicum*) pedicels [22]. The abscission of plant organs is regulated by cell wall hydrolases, with enzymes in the abscission layer increasing simultaneously with falling. Cell wall hydrolase acts on the shedding site to promote the gradual dissolution of the middle glue layer, and the primary wall of the detached cells is loosened. Finally, the cell wall is dissolved, and the organ is detached from the mother [23,24]. Polygalacturonase (PG) hydrolyzes intercellular layers and pectin substances [25], whereas cellulase (CEL) hydrolyzes cellulose [26], rendering PG and CEL important cell wall hydrolases, with an increasing trend generally observed in the activities of these enzymes during the abscission of plant organs. This phenomenon has been reported in studies investigating young citrus fruit [27], cotton (*Gossypium hirsutum*) fruit [28], *Cyclocarya paliurus* leaves [29], olive (*Olea europaea*) leaves [30], oil palm (*Elaeis guineensis*) fruit [31], apple (*Malus domestica*) fruit [32], and other plants in which organs are shed.

The shedding of plant organs is the result of gene regulation, and the shedding process is accompanied by a series of gene expression changes [33], including altered expression of hormone-related genes in the coding region and changes in the expression of genes encoding cell wall hydrolases. Transcriptome sequencing technology provides a new research tool and platform for biological science research, with transcriptomics able to analyze the differentially expressed genes (DEGs) of species in different environments, different tissues, and organs, or different physiological states; thus, this technology plays an important role in revealing the molecular mechanism of biological response, growth, and development [34,35]. Several transcriptome studies investigating plant organ abscission have been reported in recent years. For example, in a transcriptome study of cotton boll shedding [36], DEGs were found to be mainly related to plant hormone signal transduction pathways, with Eth and ABA signal transduction genes upregulated and most IAA signal transduction genes downregulated. Two gene modules related to abscission were identified using Weighted Gene Co-Expression Network Analysis (WGCNA), and genes such as *KMD3*, *TRFL6,* and *REV* in the modules were identified as hub genes. Transcriptome research into *Prunus amygdalus* [37] fruit drop indicated DEGs that were mainly enriched in biological functions such as carbon metabolism and plant hormone signal transduction, with the expression levels of the *glycosyltransferase such as family 2*, *MYB39*, *IAA13*, *gibberellin-regulated protein 11-like*, and *POD44* showing significant changes during fruit abscission. A total of 11,976 DEGs were identified in transcriptome research into *Carya illinoinensis* fruit drop [38], and the upregulation of the Eth signal transduction-related genes (*ethylene-responsive transcription factor 1B*, *EIN3-binding F-box protein 1*, *ethylene receptor 2*, and *mitogen-activated protein kinase*) and downregulation of IAA signal-related genes (*auxin-responsive protein IAA16-like*, *auxin-responsive protein IAA9-like*, *auxin response factor 9*, *auxin response factor 19*, and *auxin-induced protein X15*) indicated that hormone signaling-related genes play an important role in the observed fruit drop in pecans. Although there have been many reports on the shedding of different plant organs, few studies have investigated the shedding of calyx tubes in the Korla Fragrant Pear, and such research is thus of great significance in omics research into calyx tube shedding.

In this study, two fragrant pear varieties were selected: one a high-calyx-removal-rate variety (‘Xinnonglinxiang’) that was developed in previous research and the other a low-calyx-removal-rate-variety (Korla fragrant pear). To provide a theoretical basis for the screening of candidate genes for calyx shedding and analyze the physiological and molecular mechanisms of shedding, physiological, cytological, and transcriptomic analyses of calyx shedding were carried out in the two varieties at different shedding stages. This study is expected to provide a scientific basis for the cultivation of fragrant pear varieties with high calyx removal rates.

## 2. Results

### 2.1. Calyx Tube Shedding Rate Statistics

Statistics describing the calyx removal rates for the YB and PT varieties at each stage according to the 2023 field survey are shown in Figure 1. Calyx tubes began to drop for both varieties on 21 April; however, large numbers of calyx tubes started to fall during the first calyx shedding peak on 23 April, and a second calyx removal peak was also observed in both varieties on 25 April, after which the calyx tubes stopped falling, and the cumulative calyx shedding rate remained the same (Figure 1A). Final cumulative calyx shedding rates of 62.09% and 33.93% were observed for YB and PT, respectively, with the difference between the two varieties being significant (*p* < 0.05) (Figure 1B). In summary, two calyx shedding peaks and similar shedding periods were observed in both varieties; however, the cumulative calyx shedding differed considerably in the two varieties. Based on this, 21 April was preliminarily determined as the initial calyx shedding stage for both varieties, with a middle stage on 23 April and an end stage on 25 April (Figure 1A). These key periods in which calyx shedding was observed in the two varieties may thus be the key time at which the genes related to calyx shedding can be identified.

### 2.2. Determination of Physiological Indexes of the Calyx Tube Abscission Zone

Based on the statistical results of the calyx shedding rate (Figure 1), the physiological indices of the calyx abscission zone were also measured (Figure 2), with results showing that the four indices Eth content, GA_3_ content, CEL activity, and PG activity first increased and then decreased in both varieties over the period 21 April to 25 April (Figure 2D–G). These results indicate that these four indices may promote the shedding of the calyx tube in both varieties. The contents and activities of these four indices were significantly or extremely significantly higher in the calyx tube abscission area of the YB variety than in that of the PT variety (*p* < 0.05, *p* < 0.01). In addition, a downward trend was observed in the zeatin riboside (tZR) content of both varieties from 21 April to 25 April (Figure 2A), indicating an inhibitory effect for the tZR index on calyx tube shedding, while the ABA content showed a significant upward trend (Figure 2B), indicating that the ABA index may promote shedding of the calyx tube. Extremely significant differences (*p* < 0.01) were observed for the two index values tZR and ABA in the calyx abscission zones of both varieties from 21 April to 25 April, with the content in the calyx abscission zone of the YB variety significantly higher than that of the PT variety (*p* < 0.01) (Figure 2A,B). We also observed a decreasing trend followed by an increase in the IAA content for both varieties over the period 21 April to 25 April (Figure 2C), indicating that the IAA index may also have an inhibitory effect on calyx shedding.

In summary, significant or extremely significant differences (*p* < 0.05, *p* < 0.01) were observed in the seven studied physiological index values for the two varieties over the period 21 April, 23 April, and 25 April (Figure 2), indicating that the difference in the calyx shedding rate between the two varieties may be due to differences in these index values at the physiological level. However, the main physiological indices that affect calyx shedding still require elucidation. Based on the results of the calyx shedding rate survey (Figure 1) and the physiological index determination (Figure 2), 21 April was determined to be the initial shedding stage for both varieties, with 23 April as the mid-shedding stage and 25 April as the end-shedding stage. Abscission layer cell observation sampling analysis and transcriptome sequencing sampling analyses were performed based on these results.

### 2.3. Observation of Calyx Tube Abscission Zone Tissue Structure

Abscission of plant organs occurs due to the abscission layer cells produced at the abscission site, which undergo obvious structural changes during the process [9,10]. Therefore, we examined the cell morphology of the abscission zones on the calyx tubes in the two varieties at the different shedding stages (Figure 3). The results showed no abscission layer cells in the calyx tubes for either variety on April 20 and no significant difference in the morphologies of the calyx tube cells as compared to those surrounding the calyx (Figure 3A1–A4,a1–a4). A detachment zone began to form in the calyx tube on 21 April, when a small number of detached cells were observed (Figure 3B1–B4,b1–b4), and a clear and continuous detachment zone in the calyx tube on 23 April was accompanied by a large number of detached cells forming within the zone. These detached cells were composed of several layers of cells that were densely arranged, with a dense cytoplasm and small intercellular spaces, and significantly lower cell volumes than the surrounding cells (Figure 3C1–C4,c1–c4). The abscission layer cells in the calyx tube abscission zone began to break on 25 April, with a higher degree of rupture observed in the YB variety as compared to the PT variety (Figure 3D1–D4,d1–d4). The calyx tubes fell from both varieties on April 27, with a smooth and neat protective layer observed at the shedding site (Figure 3E1–E4,e1–e4). In summary, obvious cell morphological changes were observed at the different shedding stages of the calyx tube in both varieties, indicating that the formation of the abscission zone and the active changes in the cells comprising the abscission layer led to the shedding of the calyx tube.

### 2.4. Transcriptome Sequencing Analysis of the Calyx Tube Abscission Zone

#### 2.4.1. Sequencing Data Statistics

Transcriptome sequencing analysis of the abscission zones for the two varieties of calyx tubes was then performed at the different abscission stages. High-quality data were obtained from the original data using the quality control process described in Appendix A, with Q30 bases of 91.93% and above and GC values of between 45.92% and 47.19% obtained for each sample, indicating that the transcriptome sequencing data were of good quality and suitable for subsequent analyses. Mapped reads were obtained by comparing the sequencing data with the reference genome, with more than 90.03% mapped reads in each sample and a unique matching rate of 91.23%, indicating the suitability of the selected reference genome. The Pearson correlation coefficient was then calculated to study the correlation between the samples (Appendix A), with results indicating higher R^2^ values for samples in the same group as compared to the R^2^ value for samples in different groups, suggesting good repeatability for the same group of samples and poor correlation between the different sample groups. These results were used in the subsequent analysis.

#### 2.4.2. Analysis of Differentially Expressed Genes

To identify the genes that are related to calyx shedding, the two varieties were compared in the early, middle, and late stages of calyx shedding, with results showing 14,611 DEGs for YB2vsPT2 with 6787 upregulated genes and 7824 downregulated genes in the early shedding stage; 18,680 DEGs for YB4vsPT4 with 8599 upregulated genes and 10,081 downregulated genes during the middle stage; and 3055 DEGs for YB6vsPT6 with 1427 upregulated genes and 1628 downregulated genes at the end of abscission (Figure 4). The highest number of DEGs was therefore observed in the middle of shedding, with the obvious differences observed at this stage suggesting dramatic changes at the molecular level. The comparative combination of large numbers of DEGs is obviously more meaningful for data mining. The results obtained indicate that the mid-shedding stage may be a critical period during which the calyx removal rates are affected in both varieties.

#### 2.4.3. GO Enrichment Analysis of Differentially Expressed Genes

After obtaining the DEGs, further gene functional annotation was performed for both varieties in the three study periods using GO enrichment analysis on the 20 most prevalent DEGS. Terms with the most significant enrichment results from each comparison combination were then used to draw a common enrichment map (Figure 5). The results showed that the common functional DEG pathways were microtubule-based movement, microtubule-based processes, movement of cell subcellular components, motor activity, microtubule binding, and tubulin binding and that the proportion of DEGs in these common functional pathways changed during the different shedding periods in both varieties. The number of DEGs was significantly higher in the middle stage than in the early or late stages of shedding. These functional pathways are closely related to the composition of the cell wall and changes in the cell wall components, and the shedding of plant organs is caused by active changes in the cell wall. These enriched pathways indicate that changes in the cell wall-related genes may be related to calyx shedding in both varieties.

#### 2.4.4. Kyoto Encyclopedia of Genes and Genomes (KEGG) Enrichment Analysis of Differentially Expressed Genes

To understand the functional role of genes related to calyx shedding in different metabolic pathways, KEGG enrichment analysis was performed for the DEGs between the two varieties over the three periods, and the most significant first 20 pathways from the enrichment results for each comparison group were selected to draw a common enrichment map (Figure 6). The results showed that the common metabolic pathways of the DEGs in different calyx shedding periods of the two varieties were plant hormone signal transduction, motor proteins, glycine, serine, and threonine metabolism, and carbon metabolism. The plant hormone signal transduction pathway was the most significant, indicating that this pathway may be closely related to calyx tube shedding in both varieties.

#### 2.4.5. Analysis of Gene Expression Patterns That Are Related to the Plant Hormone Signal Transduction Metabolic Pathways

According to the results of the KEGG enrichment analysis (Figure 6), the most significant DEG pathway is plant hormone signal transduction. Plant hormones regulate cell growth and development and play important roles in plant organ abscission [12]. We therefore focused on the expression trends of DEGs in this pathway, with the DEGs that were continuously expressed during calyx shedding in the two varieties considered as genes of interest. A total of 31 continuously expressed DEGs were found between the two cultivars in this pathway (Figure 7), and the expression of these DEGs was analyzed using heatmap analysis, with results showing dramatic change in the expression of the DEGs in the YB variety (Figure 7). Based on this, and to further screen the genes whose expression change trend was consistent with the change trend of the physiological index value, the DEGs with obvious expression change trends were divided into two categories: (1) high expression in Y4, with significant upregulation as compared to Y2 and significant downregulation in Y6; and (2) low expression in Y4, with significant downregulation as compared to Y2, and significant upregulation in Y6.

Based on this classification, 14 genes are category 1: *auxin transporter-like protein* (LOC103931178), *auxin-responsive protein* (LOC108868455, LOC103939002), *ethylene receptor* (LOC103967492), *gibberellin receptor* (LOC103928345, LOC103938790), *EIN3-binding F-box protein* (LOC103958329, LOC103965573), *transcription factor MYC2* (LOC125480577), *protein phosphatase 2C* (LOC103935784, LOC103943902), *basic form of pathogenesis-related protein* (LOC103938257), *BTB/POZ domain and ankyrin repeat-containing protein NPR1* (LOC103947404), and *bZIP transcription factor* (LOC103937993). The expression levels of these 14 genes were significantly higher in Y4 than P4 (Figure 7). In addition, 16 genes belong to category 2: *auxin-responsive protein* (LOC103927977, LOC103930093), *auxin-induced protein* (LOC103931291), *auxin response factor* (LOC125468890, LOC103927986), *DELLA protein* (LOC103959476), *jasmonoyl-L-amino acid synthetase* (LOC125468985), *two-component response regulator* (LOC125472953, LOC125477598, LOC125478943, LOC125479878, LOC103942470), *protein TIFY* (LOC103929049, LOC103950252), *D3-1 cyclin-D3-1* (LOC103945321), and *histidine kinase* (LOC103958138). The expression levels of these 16 genes were lower in Y4 than in P4 (Figure 7).

The above results show that the expression levels of genes related to plant hormone signal transduction and metabolic pathways differ significantly in association with the calyx shedding process of the two varieties. We speculate that this may be an important reason for the higher calyx shedding rate of the YB variety as compared to the PT variety.

#### 2.4.6. Analysis of Gene Expression Patterns Related to the Functional Pathway of Cell Wall Hydrolase Activity

Plant organ abscission is closely related to changes in the cell wall hydrolase activity [24]. CEL and PG are two important hydrolases present in the cell wall. As shown in Figure 5, only the 20 most significant terms were selected from the GO enrichment results of each comparison group to draw the common enrichment map. However, the DEGs for the different abscission stages of the calyx tube were also enriched in the cellulose synthase activity and PG activity functional pathways (Appendix A). Therefore, we focused on the changing trends in DEG expression in these two functional pathways. To further screen the genes with the same trend of expression change and physiological index value change, DEGs with obvious changes in expression were also divided into two categories, using the same division principle as that used for genes related to the plant hormone signal transduction pathway, as described in Section 2.4.5.

The results showed six continuously expressed DEGs that were shared between the two cultivars in the cellulose synthase pathway (Figure 8), with the expression heat maps obtained for each sample indicating dramatic changes in the expression of these DEGs in the YB variety (Figure 8). Based on the classification principle, two genes belonged to category 1: *Cellulose synthase-like protein* (LOC103936880 and LOC103944558), for which the expression levels were higher in Y4 than P4 (Figure 8), and four to category 2: *Cellulose synthase-like protein* (LOC103951502, LOC103959701, and LOC103956004) and *Cellulose synthase* (LOC103943043), with all four genes showing lower expression in Y4 as compared to P4 (Figure 8).

Seven continuously expressed DEGs were also shared in the PG activity functional pathway in both varieties (Figure 9), with the DEG expression heat map (Figure 9) showing dramatic change in the expression of these DEGs in the YB variety, particularly the two category 1 genes: *exopolygalacturonase* (LOC103928400) and *polygalacturonase* (LOC103938334), for which the expression levels were higher in Y4 than P4 (Figure 9). Five genes were found to belong to category 2, *polygalacturonase* (LOC103929686, LOC103936958, LOC103941345, LOC125471551, and LOC125477850), all of which showed lower expression levels in Y4 than P4 (Figure 9).

These results show significant differences in the expression of cell wall hydrolase activity-related genes for the two varieties, which may be an important reason for the higher calyx removal rate observed for the YB variety as compared to the PT variety.

#### 2.4.7. Weighted Gene Co-Expression Network Analysis

To further explore the genes related to calyx shedding, the transcriptome data of 18 samples obtained during calyx shedding were subjected to WGCNA. After filtering out gene data with an expression level of 0, the remaining 27,963 genes, of which the top 2000 genes had the largest average absolute deviation, were used to construct a weighted gene co-expression network. Because no genes that are directly related to tZR and the plant hormone signal transduction pathways were found in the results of the gene expression patterns (Figure 7), tZR was not considered a phenotypic physiological index for WGCNA. According to the literature, ABA is not directly involved in the shedding of plant organs but only induces organs to enter senescence in advance. ABA can stimulate the production of Eth, and the real shedding process is caused by Eth rather than ABA [39]. ABA was also not considered a phenotypic marker for the WGCNA in this study. Additionally, salicylic acid (SA) and jasmonic acid (JA) are related to plant resistance and defense [40]; therefore, these two physiological indicators were not considered. Five physiological indicators were selected: GA_3_, IAA, ETH, PG, and CEL, which are closely related to calyx shedding and were therefore used as phenotypic data for WGCNA.

The results of the cluster analysis (Figure 10A) showed the clustering of 2000 genes into five modules, and the analysis focused on the modules that were significantly correlated with phenotypic data. The correlation heat map between the modules and phenotypic data (Figure 10B) showed that the blue module (MEblue) containing 137 genes was significantly negatively correlated with Eth (R = −0.83, *p* < 0.01), while the yellow module (MEyellow) contained 47 genes that are also significantly negatively correlated with Eth (R = −0.67, *p* < 0.01). This result indicates that the blue and yellow modules may be the most critical modules and that the genes in these two modules may be related to calyx shedding under the Eth response.

Further determination of the core and key core genes in the module indicated 105 core genes (Figure 11A) and 31 key core genes (Figure 11B) in the blue module that are significantly negatively correlated with Eth. To further narrow the range of key core genes, the eight key core genes annotated as Eth-related or Eth-induced from the 31 key core genes (*C7317*, *PIP22*, *SAUR50*, *EIX2*, *SUS*, *ALE2*, *MA658*, *PER64*) were identified. Protein interaction network (PPI) analysis of the 105 core genes in the blue module (Figure 11C) revealed that 11 of the genes interact with each other. The use of the CytoHubba plug-in in Cytoscape for further analysis (Figure 11D) revealed five core genes with high connectivity (*IRX10*, *IRX9*, *ESK1*, *GXM1*, *GXM3*), with the highest MCC scores observed for *IRX10* and *IRX9*.

We also found thirty-nine core genes (Figure 12A) and eight key core genes (Figure 12B) in the yellow module that were significantly negatively correlated with Eth. To further narrow the range of the key core genes, two key core genes (*M3K20* and *FB316*) annotated as Eth-related or Eth-induced were found among the eight key core genes. PPI analysis of the thirty-nine core genes in the yellow module indicated nine interacting genes (Figure 12C), and further analysis with Cytoscape (Figure 12D) indicated five key core genes with high connectivity (*OXI1*, *ATL2*, *DURF2*, *NHL10*, *PUB21*), of which the highest MCC score was obtained for *OXI1*.

Combination of the blue and yellow modules showed 20 possible key core genes (Appendix A). Heat map analysis of the expression levels of these genes in each (Figure 13) indicated significant differences in the expression levels of the 20 key core genes in the calyx shedding process for the two varieties, with a more active expression trend observed for the calyx shedding process in the YB variety. We speculate that the differential expression of these 20 key core genes in the calyx abscission zone of the two varieties may be an important reason for the significant difference observed in the calyx abscission rates for the two varieties.

To verify the reliability of the key modules obtained using the WGCNA method, GO enrichment analysis of the core genes in the blue and yellow modules was conducted, with results (Appendix A) showing that the main functional pathways significantly enriched by the core genes in the blue module were xylan metabolic process (GO: 0010410), xylan metabolic process (GO: 0045491), xylan biosynthetic process (GO: 0045492), cell wall polysaccharide biosynthetic process (GO: 0070592), and cell wall polysaccharide metabolic process (GO: 0010383), while leaf senescence (GO: 0010150) and plant organ senescence (GO: 0090693) were obtained for the yellow module (Appendix A). In summary, the functional pathways enriched by core genes in the two modules were mainly concentrated on processes related to cell wall anabolism and plant organ senescence, indicating that the core genes in both modules could be significantly enriched in functional pathways with practical biological significance. These results also indicate that these functional pathways may be related to the shedding of the calyx tubes in both varieties.

#### 2.4.8. Fluorescence Quantitative Verification

To further verify the accuracy of the transcriptome data, Real-Time Fluorescence Quantitative (RT-qPCR) analysis was used to verify nine randomly selected genes from the differentially expressed and key core genes. The transcriptome expression patterns of the nine genes (Appendix A) showed that the expression levels of five genes first decreased and then increased during calyx shedding in the YB variety, whereas the PT variety showed a continuously increasing trend. The five identified genes were *cytochrome P450* (LOC103927261), *aquaporin PIP2-2* (LOC103927853), *sucrose synthase* (LOC103950289), *polygalacturonase* (LOC103941345), and *endoglucanase* (LOC103943053). Simultaneously, the expression levels of *EIN3-binding F-box protein* (LOC103965573), *transcription factor MYC2* (LOC125480577), and *protein phosphatase 2C* (LOC103935784) first increased and then decreased in the YB variety and first decreased and then increased in the PT variety. In addition, the expression of *NDR1/HIN1-like protein 3* (LOC103940668) first decreased and then increased in both varieties.

The fluorescence quantitative results (Appendix A) showed that the fluorescence quantitative expression trend of the above nine genes was consistent with the transcriptome expression, indicating high reliability for the transcriptome sequencing data. The RT-qPCR results further confirmed that the expression levels of hormone-related and cell wall hydrolase activity-related genes changed significantly during the calyx shedding process in both varieties, with the expression trend more obvious in the YB variety.

## 3. Discussion

### 3.1. Differences in Gene Expression Patterns for the Plant Hormone Signal Transduction Pathway and Cell Wall Hydrolase Activity in the Two Varieties

Plant organ abscission typically occurs within the abscission zone [41], with changes in the various endogenous hormones and cell wall hydrolase activities in the abscission zone leading to organ shedding [42,43]. The content of endogenous hormones and the activity of cell wall hydrolases are regulated by related genes [44], and hormones are primarily involved in plant activities through signal transduction [45]. The most fundamental reason for plant organ abscission is the enhanced activity of cell wall hydrolase in the abscission zone, which promotes the degradation of the cell wall and leads to abscission [46]. The results of the gene expression patterns related to plant hormone signal transduction pathways in the two varieties (Figure 7) showed that the related genes were mainly annotated to the signal transduction pathways of Eth, ABA, IAA, GA_3_, SA, and JA, with no genes related to tZR. This indicates that tZR is not the main endogenous hormone affecting calyx shedding in either of the two varieties. Few studies have investigated the shedding of plant organs as a result of zeatin, which is only found in association with calyx shedding in Korla Fragrant Pear, for which high zeatin content in the lower part of young Korla fragrant pears was found to inhibit shedding of the calyx tube [47]. We therefore chose to determine this index to explore the changing trend of zeatin content in association with calyx shedding in the ‘Xinnonglinxiang’ variety. Interestingly, we found that the tZR index might not be directly related to the regulation of calyx shedding. However, the expression levels of the six hormone signal transduction pathways and two cell wall hydrolase activity-related genes observed in the two varieties differed significantly during calyx shedding. We speculate that this may be an important reason for the high calyx removal rate in the YB variety. A summary of the possible mechanisms of calyx shedding at the transcriptome level was produced based on these results (Figure 14).

Our prediction of the mechanism model states that the abscission of plant organs is due to the degradation of the cell wall in the abscission zone, and the degradation of the cell wall is related to the content of endogenous hormones and the activity of cell wall hydrolase in the zone. Hormone and enzyme signals are transmitted to the abscission zone as abscission signals, and the abscission zone senses the abscission signals and regulates the expression of specific hormone- and enzyme-related genes. This results in significant differences in the expression levels of the genes that are related to the six hormone signal transduction pathways and the activity of the two cell wall hydrolases. These genes were partially upregulated and partially downregulated during calyx shedding in both varieties, with an active trend observed in the YB variety. Simultaneously, the abscission layer of the cells continued to grow and develop. Initiation of the abscission process leads to the abscission layer cells finally breaking, the calyx tube completing abscission, and the young fruit developing into an abscission calyx fruit (Figure 14). The calyx shedding rates differed significantly in the two varieties. The mechanism diagram, if not complete, is at least partially reliable.

### 3.2. Differences in the Expression Patterns of the 20 Key Core Genes in the Two Varieties

Transcriptome sequencing technology plays an important role in exploring DEGs in specific physiological states [48]. However, it may be difficult to identify key core genes from large amounts of data. The WGCNA method is an effective method for identifying key core genes, dividing genes with similar expression patterns into different modules after clustering, and performing correlation analysis between modules and phenotypic data to find the module with the highest correlation with phenotypic data. The module is the key module, and the genes in the module are key core genes [49]. We found that these two modules were highly correlated with Eth, indicating that Eth may be a key core hormone in the calyx shedding process (Figure 10). A total of 20 possible key core genes were observed (Appendix A). As discussed in terms of the genes that are related to cell wall hydrolase activity, the cell wall is closely associated with plant organ abscission. Plant cell walls are mainly composed of polysaccharides (cellulose, hemicellulose, and pectin), proteins, lignin, and other components, and the synthesis and degradation of cell wall polysaccharides are mainly regulated by a variety of enzymes and enzyme complexes [50]. Interestingly, we found several genes that are related to cell wall composition among the 20 key core genes: aquaporin *PIP22*, sucrose synthase *SUS*, microtubule-associated protein *MA658*, peroxidase *PER64*, beta-xylosyltransferase *IRX10*, beta-xylosyltransferase *IRX9*, xylan acetyltransferase *ESK1*, glucuronoxylan methyltransferase *GXM1*, and glucuronoxylan methyltransferase *GXM3*. These genes were expressed at low levels in Y4 (mid-shedding), with significant downregulation as compared to Y2 (early shedding) and a lower expression level in Y4 as compared to P4 (Figure 13). Notably, existing studies on plant organ abscission have shown that Eth is closely related to plant organ abscission. Ethanol enhances the activity of cell wall hydrolases in the abscission zone by inducing their synthesis, after which the structure of the cell wall and matrix gradually loosens and the cells in the abscission zone gradually decompose, eventually leading to organ shedding [51,52]. This has been confirmed in studies on litchi [53] and citrus fruits (*Citrus reticulata*) [54]. This finding is similar to the results of the present study. This further illustrated that the key core genes obtained using the WGCNA method were reliable.

Of the 20 key core genes, the gene with the highest MCC score was considered to play the most important role, and the highest MCC scores were obtained for beta-xylosyltransferase *IRX10*, beta-xylosyltransferase *IRX9*, and serine/threonine-protein kinase *OXI1* (Figure 11D and Figure 12D). β-xylosyltransferase can promote the synthesis of xylan, which is the main component of the plant cell wall [55] and is believed to enhance cell wall strength and maintain cell structure stability [56]. This result indicates that β-xylosyltransferase is involved in the formation of the plant cell wall and affects the growth and development of plants. Interestingly, in our results, *IRX10* and *IRX9* were expressed at low levels in Y4 (mid-shedding), with significant downregulation as compared to Y2 (early shedding), and lower expression levels were observed for Y4 as compared to P4 (Figure 13). We speculate that in the middle stage of calyx shedding, low expression of *IRX10* and *IRX9* leads to a decrease in xylan synthesis, resulting in a decrease in the composition of the cell wall in the abscission zone. The cell wall structure gradually loosens and the stability weakens, eventually leading to the shedding of the calyx tube. Serine/threonine protein kinases are closely associated with Eth, and there are two major Eth receptor subfamilies, with the *ETR2* and *ERS2* receptors in subfamily II demonstrating serine/threonine protein kinase activities [57,58]. In addition, serine/threonine protein kinases include mitogen-activated protein kinase (*MAPK*), which is involved in the Eth signaling pathway [59]. For example, one study investigating the *MAPK* activity in *Arabidopsis thaliana* found that leaves treated with exogenous ETH produced proteins similar to *MAPK,* indicating upregulation in the expression of the *MAPK*-related signal genes and confirming the involvement of *MAPK* in the signal transduction pathway of Eth [60]. Another study of Eth biosynthesis in Arabidopsis thaliana found that *MAPK* was involved in the biosynthesis of Eth by regulating the expression of the key enzyme gene *ACS* at the transcriptional level [61]. Interestingly, we also found an MAPK gene, *M3K20*, among the 20 key core genes, with *OXI1* and *M3K20* expressed at low levels in Y4 (mid-shedding) and highly expressed in Y6 (late shedding). We therefore speculate that at the end of calyx shedding, the high expression of *OXI1* and *M3K20* enhances the activity of serine/threonine protein kinase, which indirectly leads to enhancement of the Eth receptor activity and an increase in the endogenous Eth synthesis, promoting calyx shedding.

Based on these results, a possible calyx shedding mechanism at the WGCNA level is summarized in Figure 15. Our prediction of the mechanism model is that under natural conditions, miRNAs mediate the shedding of the calyx tube by regulating the expression of Eth synthesis-related or Eth induction-related genes. Under the regulation of the 20 key core genes, the Eth content in the calyx abscission zone of the two varieties changes significantly, and with the further growth and development of the detached cells, the abscission zone begins to break and the calyx tube gradually falls, with the different gene expression eventually leading to the significant differences observed in the calyx removal rates of the two varieties (*p* < 0.05). The mechanism map produced using this information integrates the molecular, physiological, cytological, and phenotypic trait levels (Figure 15) and lays a theoretical foundation for further studies on the molecular mechanism underlying the high calyx removal rate in the YB variety. We also verified some randomly selected genes from the key core genes using RT-qPCR (Appendix A). These results indicate that the mechanism diagram is reliable, if not complete, at least in part.

## 4. Materials and Methods

### 4.1. Experimental Materials

The experimental site is located at the Luntai County Horticultural Field (41°05′–42°32′ N, 83°38′–85°25′ E) in the Bayinguoleng Mongolian Autonomous Prefecture, Xinjiang. The average annual temperature is 10.6 °C, average annual precipitation is 52 mm, the average annual evaporation is 2072 mm, the average annual sunshine hours is 2783 h, and average annual total solar radiation is 577.6 kJ/cm^2^. Twenty trees of the Korla fragrant pear bud mutation variety ‘Xinnonglinxiang’ with the same growth and no pests or diseases and under the same management measures and site conditions were selected as the YB group, and 20 Korla fragrant pear trees were selected as the CK group. ‘Xinnonglinxiang’ is a bud mutation variety of the Korla fragrant pear that shows specific changes in fruit calyx removal under natural pollination conditions and is confirmed to be a stable genetic variation that has been cultivated via grafting for many years. This study was approved by the Xinjiang Uyghur Autonomous Region Forest Variety Approval Committee in 2023. The bud, flowering, leaf growth, new shoot growth, and fruit growth periods of ‘Xinnonglinxiang’ and Korla fragrant pears are basically coincident, and the phenological period is basically the same. No significant difference in fruit quality is observed between the two groups (*p* > 0.05); however, there is a significant difference in the fruit calyx removal rate, with the fruit calyx removal rate of ‘Xinnonglinxiang’ significantly higher than that of the Korla fragrant pear (*p* < 0.05) [62]. The studied trees from both ‘Xinnonglinxiang’ and Korla fragrant pear were 10 years of age, and the row spacing of the plants in the garden is 4 m × 5 m. Photos of the Korla fragrant and ‘Xinnonglinxiang’ trees are provided in Figure 16. The Dangshan pear (*Pyrus bretschneideri* Rehd) is used as a natural pollinator in the garden, with the ratio of the main varieties to the pollination tree varieties at 8:1.

### 4.2. Experimental Methods

The logical framework and flow chart of the research is provided in Appendix A.

#### 4.2.1. Investigation of Calyx Tube Shedding Characteristics

Five trees with the same growth, no pests or diseases, no management measures, and the same site conditions were selected as sample trees for the YB and CK groups. Trees were selected in 2023, with two branches with the same degree of development and growth potential selected from the east, south, west, and north of each tree crown for listing and marking. After fruit setting, the number of fruits and number of calyx-off fruits on all marked fruit branches in the two groups were observed daily, and the dates on which the calyx fell were recorded until the phenomenon ceased. The standard of the calyx fruit was determined from the discoloration of the calyx tube, and a clear yellow-green boundary was observed in the calyx tube in the late stage of development (Figure 17). The initial, middle, and end stages of calyx shedding in the two groups were determined based on dynamic calyx shedding. The calyx shedding rate was calculated as the ratio of calyx-shed fruits to the number of fruit settings, and the average calyx removal rate for all marked fruiting branches was determined as the calyx removal rate for each group.

#### 4.2.2. Paraffin Section Observation of Calyx Tube Detachment Cells

Synchronous with the experiment described in Section 4.2.1, three trees with the same fixed growth, no pests or diseases, management measures, and site conditions in the YB and CK groups were used as sample trees for observation of the calyx tube detachment cells. The same fruit branch marking method as that described in Section 4.2.1 was used, and after fruit setting, the abscission area of the calyx tube was collected daily from both groups from the early stage to the end of calyx tube shedding (Figure 17). Five samples were collected daily and stored in FAA fixative for paraffin section observation experiments. The FAA fixative was configured using a ratio of 5 mL glacial acetic acid, 90 mL 70% alcohol, 5 mL glycerol, and 5 mL formaldehyde [63,64]. The prepared slices were stained with the safranine-fast green staining method, and neutral gum was added to seal the slices before drying at 37 °C in an electrothermal constant-temperature blast drying box [65,66]. An optical microscope (Leica DM6000B; Leica Microsystems, Shanghai Trading Co. Ltd., Shanghai, China) was used to observe changes in the abscission layer cells during the different shedding periods.

#### 4.2.3. Determination of Physiological Indexes for the Calyx Tube Abscission Zone

To determine the physiological indexes of the trees, samples were obtained from trees using the same method as that described in Section 4.2.2, with the same fruiting branch labeling method as that described in Section 4.2.1. After self-fruiting, 20 calyx tube abscission tissue samples were collected daily from both groups during the calyx tube abscission process and placed in a liquid nitrogen tank to determine physiological indicators for the abscission area, as seen in Figure 17. High-performance liquid chromatography–mass spectrometry (HPLC-MS) was used to determine the IAA, GA_3_, ABA, tZR, and other indicators [67]. GA_3_, IAA, tZR, and ABA were used as the endogenous hormone standards (purity > 96%; Dr. Ehrenstorfer GmbH, Shanghai Zerui Biotechnology Co., Ltd. Shanghai, China). Acetonitrile, methanol, formic acid, and anhydrous ethanol were of chromatographic grade (≥98%, HPLC). An appropriate amount of the standard substance was then weighed and a stock solution (100 mg/L) was prepared with methanol, which was then stored in a refrigerator at −4 °C. Based on the requirements of the test, the standard stock solution was diluted with methanol, and a mixed standard solution was prepared at an appropriate mass concentration. An LC-MS (Agilent 1290-6460, Suzhou Krewster Measurement and Control Technology Co., Ltd. Suzhou, China) and chromatographic column ZORBAX Eclipse Plus C_18_ (rapid resolution HD 2.1 mm × 150 mm, 1.8 μm) were used for analysis under chromatographic (LC) conditions of a detection wavelength of 290 nm, column temperature of 35 °C, flow rate of 0.3 mL/min and a mobile phase, with B (methanol) and A (0.1% formic acid), and the following graduated elution: 0–1 min 10–40% (B), 1–2.5 min 40–45% (B), 2.5–5 min 45–80% (B), 5–5.1 min 80–10% (B), 5.1–7.1 min 10–10% (B) at an injection volume of 5 μL. For mass spectrometry (MS), an electrospray ionization source (ESI) in negative ion scanning mode and multi-reaction monitoring mode was utilized. The ion source temperature was 350 °C, the drying gas flow rate was 11 L/min, the nebulizer pressure was 35 psi, the capillary voltage was 4 kV, and the collision gas was high-purity nitrogen. Extraction of endogenous hormones was performed by first weighing 1.0 g of fresh calyx tube abscission zone sample and grinding it until homogenous using liquid nitrogen before placing it in a 15 mL centrifuge tube. Then, 10 mL of pre-cooled methanol-formic acid solution (99:1) was added to the centrifuge tube, and the results were subjected to ultrasonic treatment at 22 °C for 2 min and 4 °C for 12 h. The extract was then centrifuged for 10 min at 10,000 rpm, and after absorbing 1.0 mL supernatant, H_2_O was added to obtain 10 mL samples. An ODS C_18_ solid-phase extraction column was then washed twice using 6 mL of a 10% methanol solution, and the solid-phase extraction column was utilized for sample adsorption. Finally, 6 mL of a methanol–formic acid solution (99:1) was used for elution. The collected eluent was decompressed and concentrated to a small volume before diluting to 1.0 mL with methanol and filtering using a 0.22 μm microporous membrane. The solution was then used in the detection of endogenous hormones. The Eth content was determined using gas chromatography–mass spectrometry (GC-MS) [68], for which fresh calyx tubes (10 g) were weighed and sealed in wide-mouthed bottles with a sealing film, and left in the dark for 12 h and the Eth content determined using GC-MS (Agilent 7890A/5975C, Beijing JingKorida Technology Co., Ltd. Beijing, China) with a chromatographic column: HP-55% (capillary column), phenyl methyl siloxane capillary 30.0 m × 320 μm × 0.25 μm nominal, and a hydrogen ion flame detector (FID) with a column temperature of 100 °C and a detector temperature of 150 °C. The flow rate of the N_2_ carrier gas was 50 mL·min^−1^, the flow rate of the H_2_ gas was 50 mL·min^−1^, the air flow rate was 400 mL·min^−1^, and the retention time was 2.5 min. An injection needle was then used to remove 25 μL of the Eth sample before determination on the chromatograph. The gas sample was injected at a flow rate of 1.5 mL·min^−1^, and the determination was repeated thrice. The PG activity was determined using the DNS colorimetric method [69], and the activity of cellulose CEL was determined using anthrone colorimetry [70].

#### 4.2.4. Transcriptome Sequencing of the Calyx Tube Abscission Zone

Synchronous with the experiment described in Section 4.2.1, and using the same sample tree fixation method as that described in Section 4.2.2 and fruiting branch labeling method as that described in Section 4.2.1, 10 tissue samples were removed daily from the calyx tube separation area in both the YB and CK groups after the fruit had set and placed in a liquid nitrogen tank for quick freezing and preservation. Based on the survey results of the calyx shedding characteristics described in Section 4.2.1, two groups of samples were collected at the early, middle, and late stages of calyx shedding for transcriptome sequencing. Samples from the two varieties were collected during all three shedding periods and three biological replicates were obtained, resulting in a total of 18 samples (Y2-1, Y2-2, Y2-3, Y4-1, Y4-2, Y4-3, Y6-1, Y6-2, Y6-3, P2-1, P2-2, P2-3, P4-1, P4-2, P4-3, P6-1, P6-2, P6-3), with Y2, Y4, and Y6 denoted YB, and P2, P4, and P6 denoted PT. An RNA extraction kit (Omega Bio-Tek; Shanghai Baili Biotechnology Co., Ltd., Shanghai, China) was used to extract the sample RNA, for which calyx tube samples were selected and placed in a mortar for grinding in the presence of liquid nitrogen. Samples (25–50 mg) were then placed in a pre-cooled 1.5 mL RNAase-free centrifuge tube and 500 μL of Buffer RA was quickly added for full mixing before the addition of 100 μL chloroform and 150 μL water-saturated phenol. The tubes were then shaken for 15 s and centrifuged at room temperature for 5 min with the gravitational acceleration set to 14,000 gs^−1^. The supernatant (350 μL) was then transferred to a non-enzymatic centrifuge tube (2 mL), and 350 μL of buffer RC was added for full mixing. The gDNA Filter Column was then placed in a 2 mL collection tube, and the supernatant was transferred to the column before centrifugation at room temperature for 2 min with the gravitational acceleration set to 14,000 gs^−1^. The filtrate was then extracted, 350 μL of anhydrous ethanol was added, and the pipette gun was used 5–10 times. The Hibind RNA Mini column was then inserted into a 2 mL collection tube, and the mixture was obtained. The results were then centrifuged at room temperature for 1 min, and the filtrate was discarded. The gravity acceleration was set to 10,000 g per second, and the Hibind RNA Mini column was inserted into the same 2 mL collection tube before adding 400 μL RWC Wash Buffer and centrifuging at room temperature for 1 min, with the gravity acceleration set to 10,000 gs^−1^. The filtrate was then discarded, and a Hibind RNA Mini column was inserted into the 2 mL collection tube. A 500 μL RNA Wash Buffer II was then added for dilution, and the sample was washed twice before centrifugation at room temperature for 1 min. The filtrate was then discarded, the gravity acceleration was set to 10,000 gs^−1^, and the Hibind RNA Mini column was inserted into the same 2 mL collection tube. The empty column was then centrifuged at room temperature with the gravitational acceleration set to 10,000 gs^−1^. DEPC water (30–50 μL) was then added to the center of the column, and the column was left at room temperature for 2 min and centrifuged for 1 min before the RNA was eluted with the gravity acceleration set to 10,000 gs^−1^. To improve the elution efficiency, the obtained filtrate was re-absorbed back into the column, and the RNA was eluted after standing for 2 min and centrifuging for 1 min at room temperature with the gravity acceleration set to 10,000 gs^−1^. The final extracted RNA was then placed in an ultra-low-temperature refrigerator at −80 °C for use. After RNA extraction, an Agilent 2100 Bioanalyzer (Beijing Longyue Biotechnology Development Co., Ltd. Beijing, China) was used to determine the sample RNA quality. To construct the library, the fragmented mRNA was then used as a template, and random oligonucleotides were used as primers to synthesize the first strand of cDNA using the M-MuLV reverse transcriptase system, with RNase H used to degrade the RNA strands. A second strand of cDNA was then synthesized from the dNTPs. After purification of the double-stranded cDNA, AMPure XP beads were used to screen for cDNA of approximately 250–300 bp by end repair, a tail was added, and the sequencing adapter was connected. PCR amplification was then performed, and the PCR products were purified using AMPure XP beads, allowing the construction of the library. The successfully constructed library was then sequenced using the Illumina HiSeq platform [71].

#### 4.2.5. Sequencing Data Analysis

Fastp software (https://github.com/OpenGene/fastp, accessed on 7 August 2024) was used to obtain sequencing data (reads) and filter any low-quality reads. The obtained clean data were then subjected to quality inspection using FastQC (Version 0.11.8), and the qualified data were used for subsequent analysis. HISAT2 (Version 2.0.5) [72] was utilized to align the clean reads with the reference genome for the white pear, *Pyrus x bretschneideri*, which can be found at https://ftp.ncbi.nlm.nih.gov/genomes/all/GCF/019/419/815/GCF_019419815.1_Pyrus_bretschneideri_v1/GCF_019419815.1_Pyrus_bretschneideri_v1_genomic.fna.gz, which was accessed on 7 August 2024. Meanwhile, StringTie [73] software (Version 2.2.3) was used to calculate the abundance of gene transcripts based on the alignment results and the gene expression quantified by calculating FPKM values (fragments per kilobase of exon model per million mapped fragments). The DESeq2 package [74] was then used to analyze the DEGs, with P-adjust ≤ 0.05 and |log2FC| ≥ 1 used as criteria for the screening of DEGs in the different calyx tube shedding periods. ClusterProfiler software (Version 3.8.1) was used to perform GO and KEGG enrichment analyses of the DEGs in the calyx tube abscission zone.

#### 4.2.6. Weighted Gene Co-Expression Network Analysis

A weighted gene co-expression network was constructed using R4.2.1 software with the WGCNA package. Clustering was used to divide the module, and the key module was determined by associating each module with the phenotypic data. Two index values were then calculated to determine the core genes in the key modules: (1) the correlation coefficient (MM value) between the gene expression in the module and the module characteristic value, and (2) the correlation coefficient (GS value) between the gene expression and phenotypic data in the module. A scatter plot of the MM and GS absolute values was then generated, and genes obtained with |MM value| > 0.8 and |GS value > 0.3| were assumed to be core genes [75]. To further determine the key core genes, (1) the genes in the module, core genes, and common DEGs in the three periods of the calyx tube shedding process were counted for both types of sample, and the key core genes obtained after intersecting the three categories of genes. (2) PPI analysis was then performed on the core genes in the module using the STRING database (version 12.0) to identify interacting genes. The maximum cluster centrality (MCC) was then calculated using the CytoHubba plug-in in Cytoscape (version 3.10.0) to determine genes with high connectivity [76], with genes with higher MCC scores considered the key core genes. The metascape analysis tool (https://metascape.org/gp/index.html#/main/step1 accessed 5 September 2024) was then utilized to perform GO enrichment analysis of the core genes, verifying the biological function of the core genes in the key modules obtained by WGCNA.

#### 4.2.7. Real-Time Fluorescence Quantitative Analysis

To verify the accuracy of the sequencing results, nine genes were randomly selected based on the DEGs and key core genes and subjected to RT-qPCR verification. Each process was repeated thrice, and the results were averaged. The tubulin alpha chain (TBA; gene ID: 125470975) was used as an internal reference gene, and Primer 5.0 was utilized to design the primers. The obtained sequence information is shown in Appendix A. The specific steps of the fluorescence quantitative PCR and RNA extraction method were the same as those described in Section 4.2.4, with RNA extracted and reverse transcribed using a reverse transcription reaction system (20 μL) with 5 × reaction buffer (4 μL); oligo (dT) 18 Primer (100 μM) (0.5 μL); random hexamer primer (100 μM) (0.5 μL); servicebio ^®^ RT Enzyme Mix (1 μL); total RNA * (10 μL); and rNase-free water, which was added to obtain 20 μL samples. The reverse transcription reaction system was then configured, and the solution was gently mixed and centrifuged. Reverse transcription was performed under the following conditions: 25 °C (5 min), 42 °C (30 min), and 85 °C (5 s), and quantitative PCR performed using a 0.2 mL PCR tube, with three tubes prepared for each reverse transcription product containing 2 × qPCR mix (7.5 μL); 2.5 μM gene primers (upstream + downstream) (1.5 μL); the reverse transcription product (cDNA) (2.0 μL); and rNase-free water (4.0 μL). PCR amplification was then performed with 40 cycles of pre-denaturation at 95 °C for 30 s, denaturation at 95 °C for 15 s, and annealing/extension at 60 °C for 30 s, after which the temperature was raised from 65 to 95 °C, with a fluorescence signal collected every 0.5 °C. Finally, the 2^−ΔΔCt^ method was used to calculate the relative gene expression [77].

#### 4.2.8. Data Processing

Excel (version 2018; Microsoft Corp., Redmond, WA, USA) was used for calyx abscission rate statistics and calyx tube abscission zone physiological index value statistics, and GraphPad Prism 8 software was used for mapping. Statistical significance was set at *p* < 0.05 or *p* < 0.01. The microstructure of the detached calyx cells was observed using a Leica DM6000B instrument. The statistical results of the DEGs in the calyx tube, correlation results of gene expression between samples, and heat map of the gene expression were drawn using R4.2.1. The WGCNA results were plotted using the WGCNA package in R4.2.1. The results for key core genes were drawn using Cytoscape v3.10.0. The key core gene expression heat map, module gene GO enrichment result map, and RT-qPCR result map were drawn using R4.2.1 software.

## 5. Conclusions

Exploring the molecular mechanism of the high calyx shedding rate of ‘Xinnonglinxiang’ under natural conditions is crucial for breeding varieties with a high calyx shedding rate and is important for the high-quality development of the Korla fragrant pear industry. This study is an investigation of the physiology, cytology, and molecular biology of two varieties of pears with calyx tube shedding at different rates. The results showed significant differences in the expression of genes related to the Eth, ABA, IAA, GA_3_, SA, and JA signal transduction pathways in both varieties, with the expression levels of genes related to PG and CEL activities also significantly different. The WGCNA method identified Eth as a key core hormone, and 20 key core genes were observed that are induced by Eth and may play important roles in regulating calyx shedding. Of these, *IRX10*, *IRX9,* and *OXI1* may be the hub key core genes. Finally, the possible calyx shedding mechanisms are summarized, with the produced mechanism map providing important genetic resources for the improvement of pear varieties.

## Figures and Tables

**Figure 1 plants-13-03504-f001:**
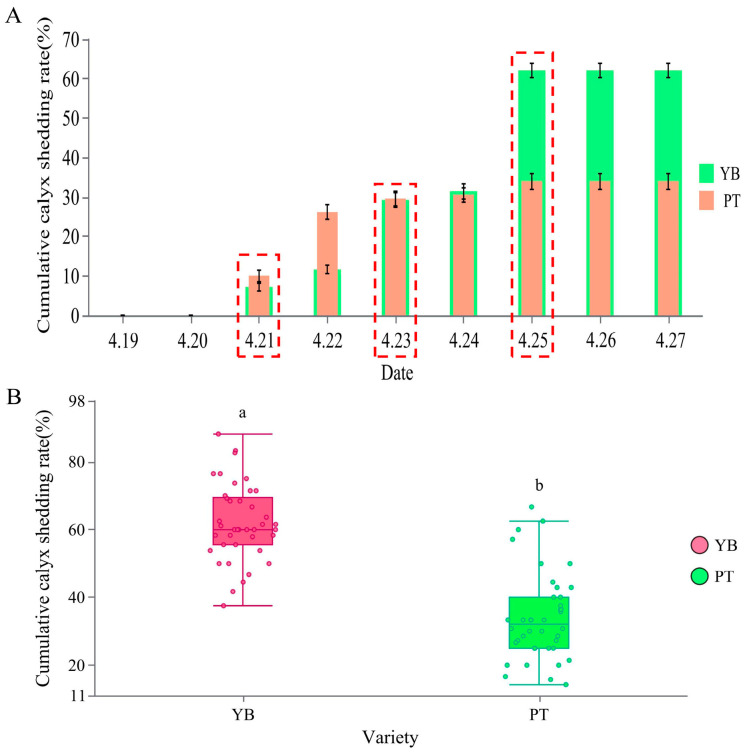
Cumulative calyx abscission rate of YB and PT at different stages. (**A**) is the statistical result of the cumulative calyx shedding rate of YB and PT, and (**B**) is the final cumulative calyx shedding rate of YB and PT. The data of figure (**A**,**B**) are expressed as mean ± SD with at least 40 biological replicates. The a and b in figure (**B**) indicate that there are significant differences between the two groups. The red dotted line frame indicates the three critical periods of calyx tube shedding.

**Figure 2 plants-13-03504-f002:**
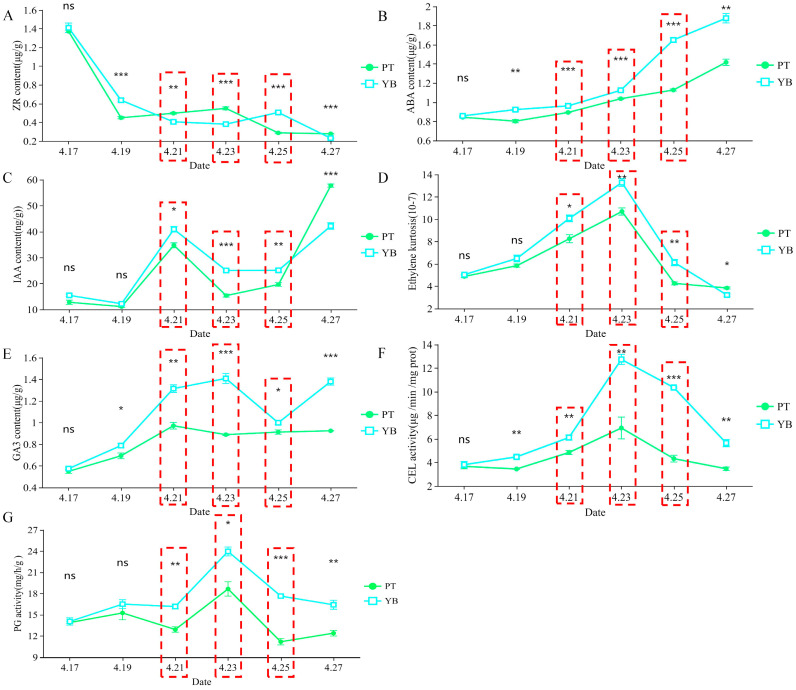
The change trend of physiological index values in the abscission zone of YB and PT variety during calyx shedding. (**A**) is zeatin (tZR), (**B**) is abscisic acid (ABA), (**C**) is auxin (IAA), (**D**) is ethylene (Eth), (**E**) is gibberellin (GA_3_), (**F**) is cellulase (CEL), (**G**) is polygalacturonase (PG). The *t*-test was used for statistical analysis, and the data are expressed as mean ± standard error of three biological replicates. The asterisk, double asterisk and triple asterisk indicate significant differences at the *p* < 0.05, *p* < 0.01 and *p* < 0.001 levels, respectively. The red dotted line frame indicates the three critical periods of calyx tube shedding. ns indicates that there is no significant difference between the two varieties.

**Figure 3 plants-13-03504-f003:**
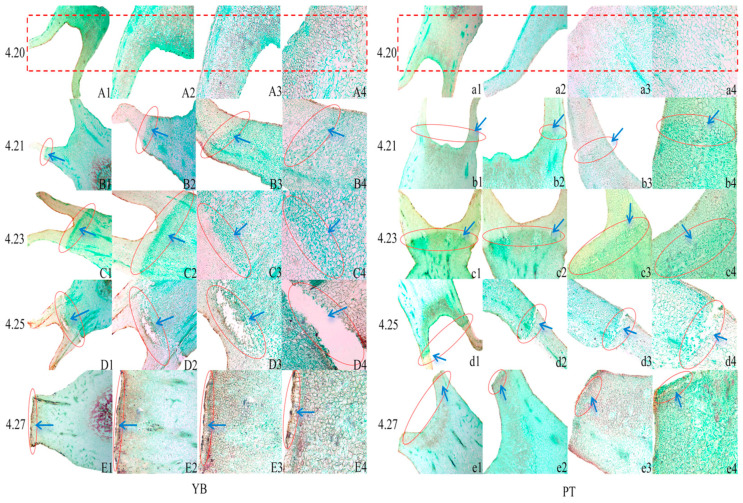
Microscopic observation of calyx tube abscission zone of YB and PT varieties. (**A1**–**E4**) are the microscopic observation results of the abscission zone of the calyx tube shedding process of the YB variety, and the left column number represents the date. (**a1**–**e4**) represent the result map of PT variety, and a column of numbers on the left represents the date. YB varieties are represented by uppercase letters. (**A1**,**B1**,**C1**,**D1**,**E1**) are images under the objective of 2.5×; (**A2**,**B2**,**C2**,**D2**,**E2**) are the pictures under 5× objective lens; (**A3**,**B3**,**C3**,**D3**,**E3**) are 10× objective lens images; (**A4**,**B4**,**C4**,**D4**,**E4**) are pictures under 20× objective lens. The PT variety is represented by lowercase letters, and the size of the picture objective is the same as that of the YB variety. The red rectangular dashed box indicates that no separation layer cells are formed. The red oval frame indicates the location of the detached cells.

**Figure 4 plants-13-03504-f004:**
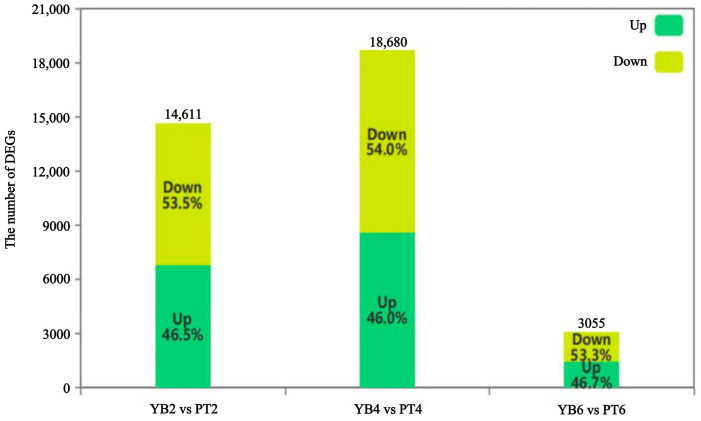
Differentially expressed genes between YB and PT at different shedding stages of calyx tube. The initial sample of calyx tube shedding (YB2, PT2); mid-shedding samples (YB4, PT4); samples at the end of shedding (YB6, PT6).

**Figure 5 plants-13-03504-f005:**
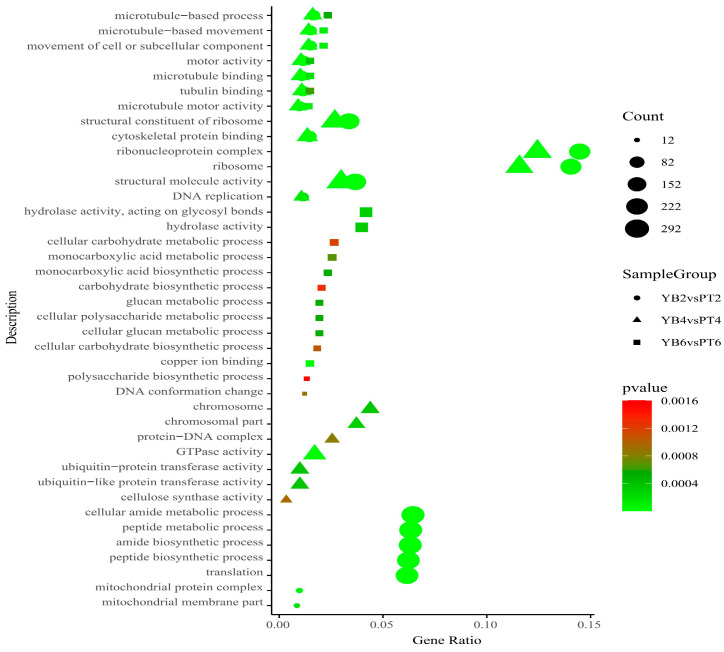
GO enrichment analysis of YB and PT differentially expressed genes at different shedding stages of calyx tube. The abscissa in the figure is the ratio of the number of differentially expressed genes enriched in the GO pathway to the total number of differentially expressed genes; the ordinate is GO enrichment pathway; the size of the point represents the number of genes enriched in the GO pathway; the color from green to red represents the significant size of the enrichment, and the closer to the green, the smaller the significance; different shapes represent different comparison combinations. The circle is YB2vsPT2 (initial shedding), the triangle is YB4vsPT4 (mid-shedding stage), and the square is YB6vsPT6 (late stage of shedding).

**Figure 6 plants-13-03504-f006:**
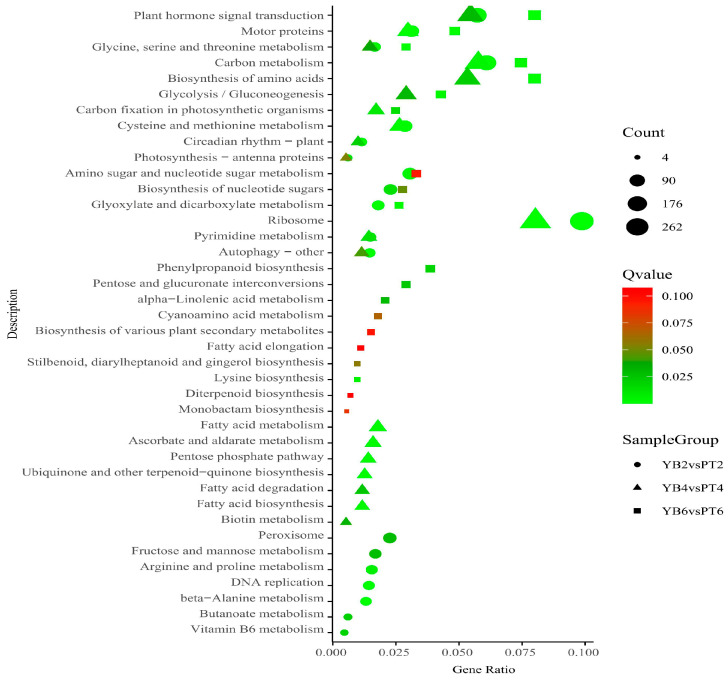
KEGG enrichment analysis of YB and PT differentially expressed genes at different abscission stages of calyx tube. The abscissa in the figure is the ratio of the number of differentially expressed genes annotated to the KEGG pathway to the total number of differentially expressed genes; the ordinate is the KEGG pathway; the size of the point represents the number of genes annotated to the KEGG pathway; the color from green to red represents the significant size of the enrichment, and the closer to the green, the smaller the significance; different shapes represent different comparison combinations. The circle is YB2vsPT2 (initial shedding), the triangle is YB4vsPT4 (mid-shedding stage), and the square is YB6vsPT6 (late stage of shedding).

**Figure 7 plants-13-03504-f007:**
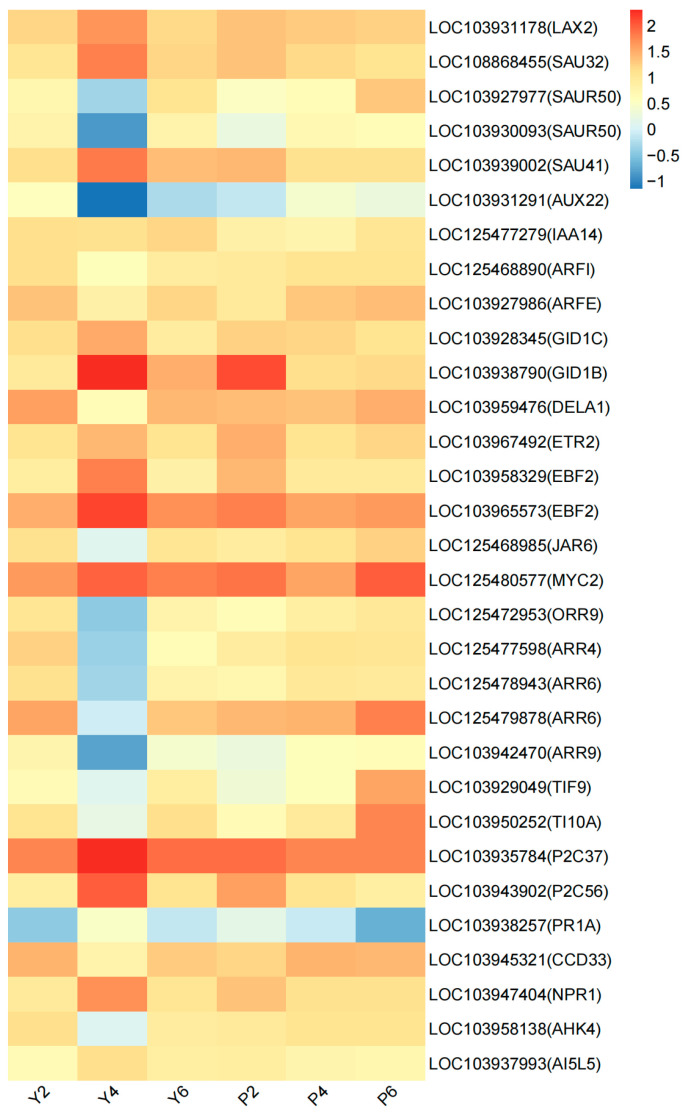
Expression patterns of genes related to plant hormone signal transduction metabolic pathway. The abscissa is the sample name, and the ordinate is the gene name. The closer the color is to red, the higher the expression is. The initial sample of calyx tube shedding (Y2, P2); mid-shedding samples (Y4, P4); samples at the end of shedding (Y6, P6).

**Figure 8 plants-13-03504-f008:**
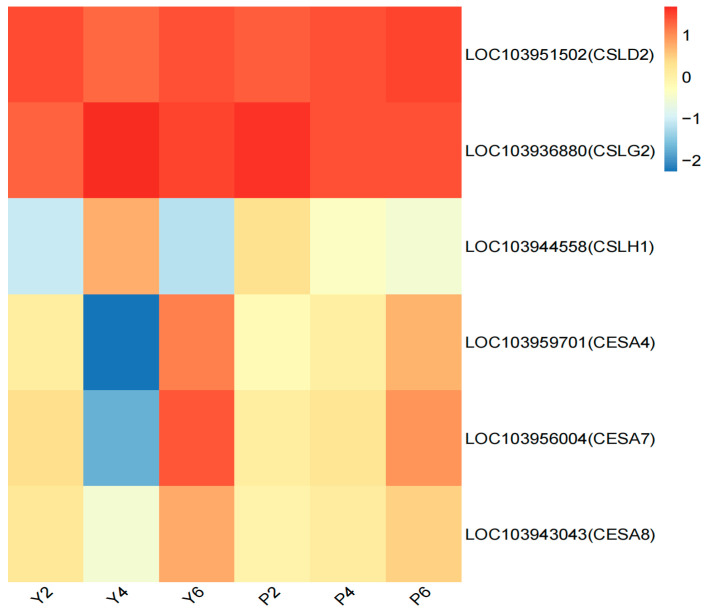
Expression patterns of genes related to the functional pathway of cellulose synthase activity. The abscissa is the sample name, and the ordinate is the gene name. The closer the color is to red, the higher the expression is. The initial sample of calyx tube shedding (Y2, P2); mid-shedding samples (Y4, P4); samples at the end of shedding (Y6, P6).

**Figure 9 plants-13-03504-f009:**
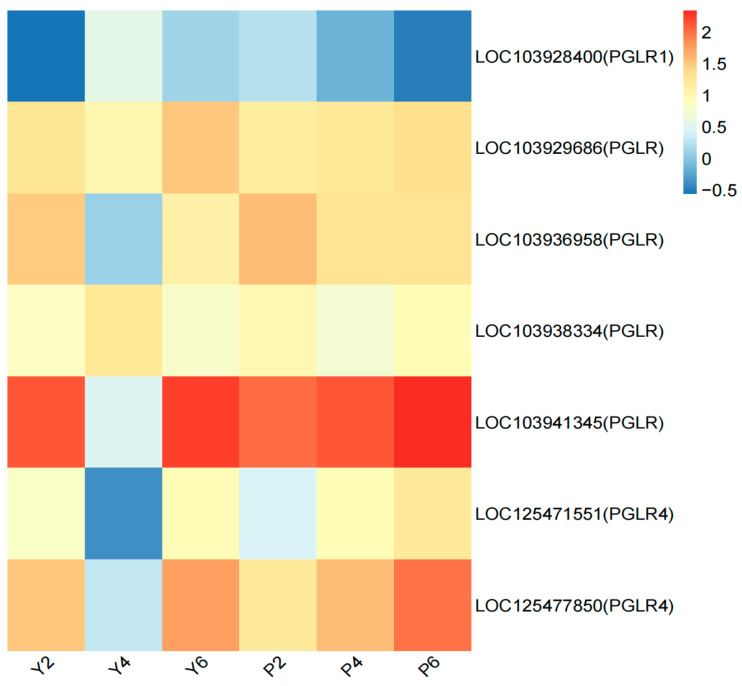
Expression patterns of genes related to polygalacturonase activity functional pathway. The abscissa is the sample name, and the ordinate is the gene name. The closer the color is to red, the higher the expression is. The initial sample of calyx tube shedding (Y2, P2); mid-shedding samples (Y4, P4); samples at the end of shedding (Y6, P6).

**Figure 10 plants-13-03504-f010:**
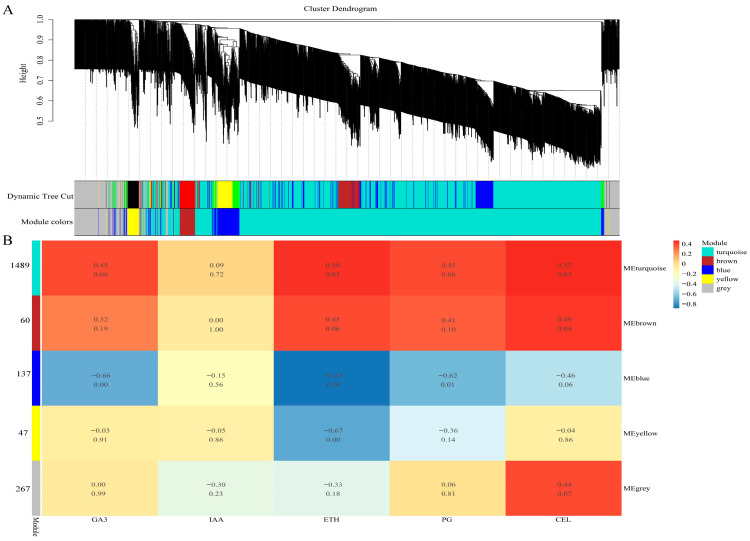
WGCNA network construction and key co-expression module identification. (**A**) is the gene clustering tree, and different colors represent different clustering modules. (**B**) is the heat map of the correlation between the clustering module and the phenotypic physiological index. The upper data in the module are the correlation coefficient R value, and the lower data are the *p*-value. The deeper the color, the stronger the correlation. The smaller the *p*-value, the stronger the significance.

**Figure 11 plants-13-03504-f011:**
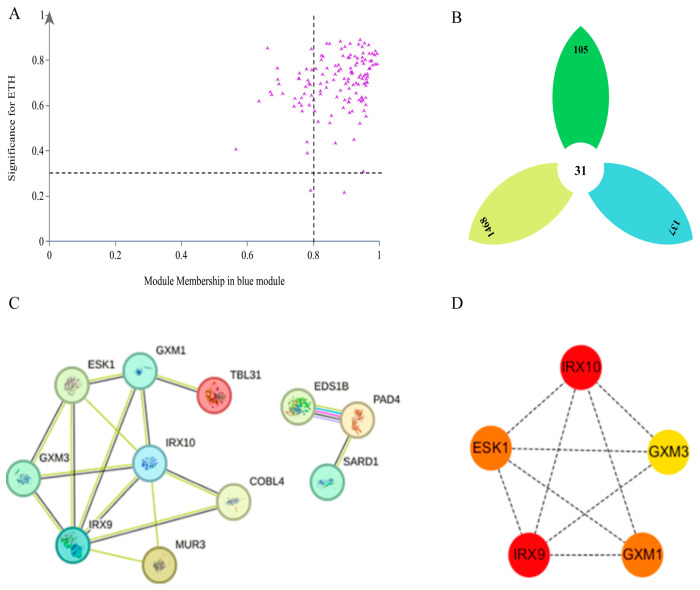
Screening of core genes and key core genes in the blue module. (**A**) is the scatter plot of the absolute values of MM and GS in the blue module, and the genes in the upper right corner region are the core genes in the module. (**B**) is the petal Venn diagram, the green petal is the core gene set in the blue module, the yellow petal is the gene set that is differentially expressed in three periods during the calyx shedding process of the two varieties, and the blue petal is the gene set in the blue module. (**C**) is the PPI interaction network diagram of the core gene in the blue module. (**D**) is the first five key core genes after further screening by Cytoscape software. The higher the MCC score, the closer the color is to red, and the more likely it is to be the key core gene in the blue module.

**Figure 12 plants-13-03504-f012:**
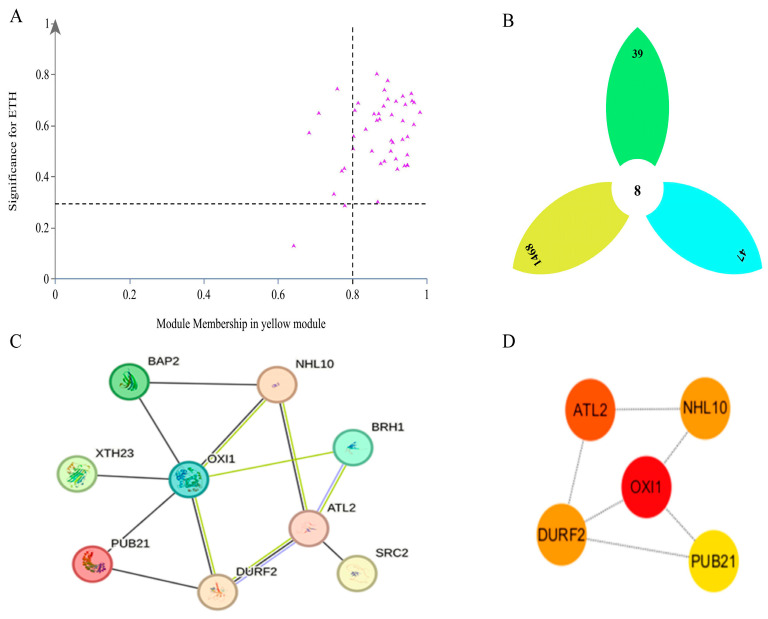
Screening of core genes and key core genes in yellow module. (**A**) is the scatter plot of the absolute values of MM and GS in the yellow module, and the genes in the upper right corner region are the core genes in the module. (**B**) is the petal Venn diagram, the green petal is the core gene set in the yellow module, the yellow petal is the gene set that is differentially expressed in three periods during the calyx shedding process of the two varieties, and the blue petal is the gene set in the yellow module. (**C**) is the PPI interaction network diagram of the core gene in the yellow module. (**D**) is the first five key core genes after further screening by Cytoscape software. The higher the MCC score, the closer the color is to red, and the more likely it is to be the key core gene in the yellow module.

**Figure 13 plants-13-03504-f013:**
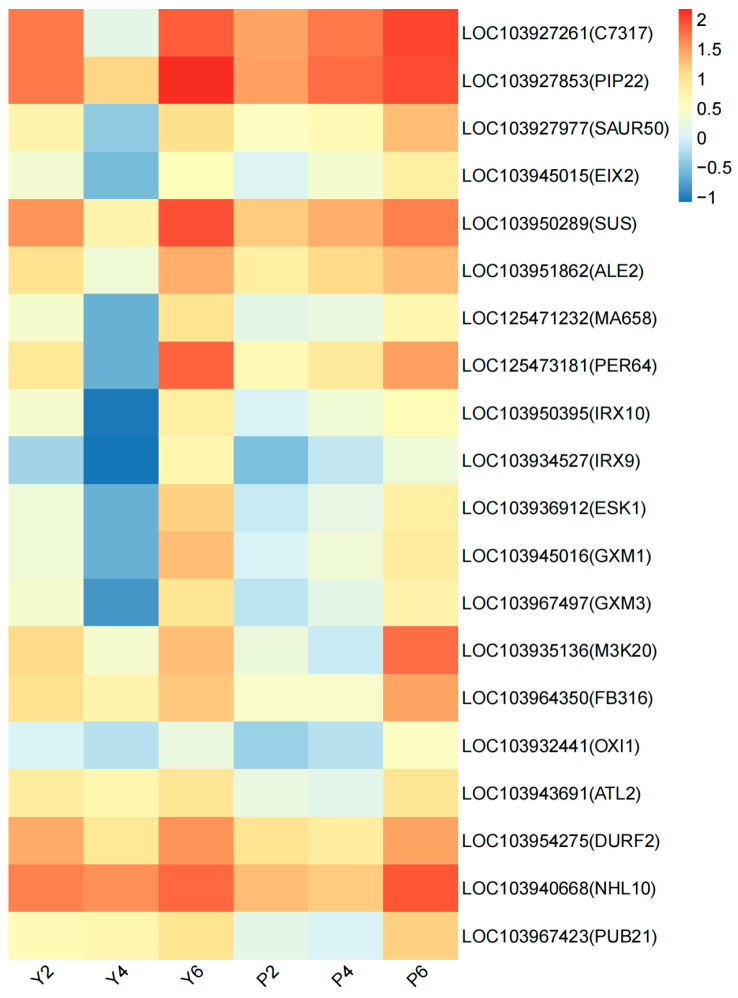
Expression heat map of 20 key core genes. The abscissa is the sample name, and the ordinate is the gene name. The closer the color is to red, the higher the expression is. The initial sample of calyx tube shedding (Y2, P2); mid-shedding samples (Y4, P4); samples at the end of shedding (Y6, P6).

**Figure 14 plants-13-03504-f014:**
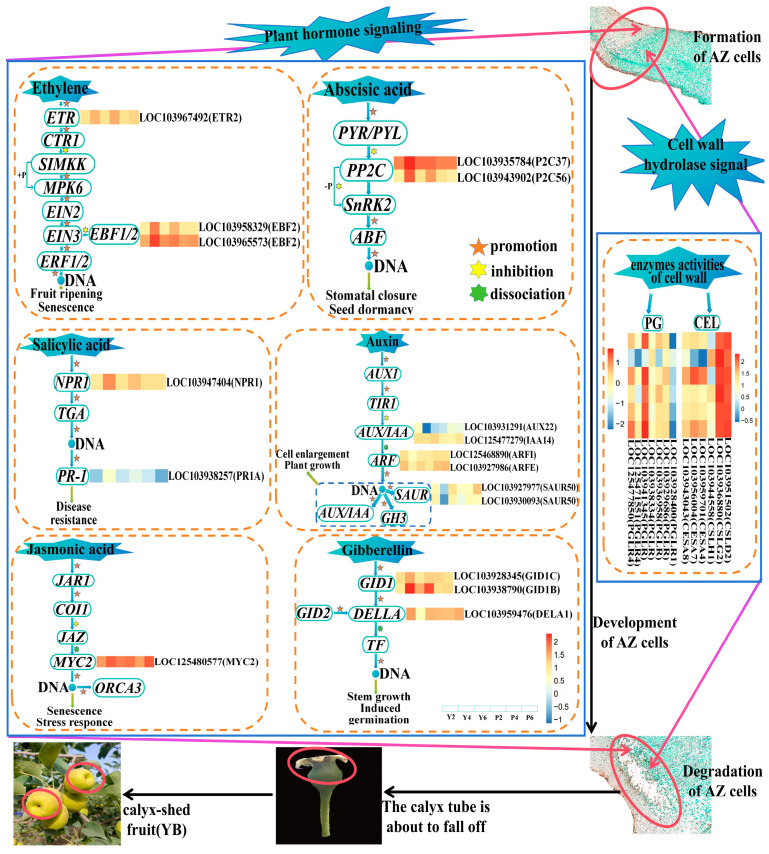
Mechanism diagram of calyx tube shedding based on transcriptomics. In the six hormone signal transduction pathways and two cell wall hydrolase activity functional pathways, the closer the color is to red, the higher the gene expression is.

**Figure 15 plants-13-03504-f015:**
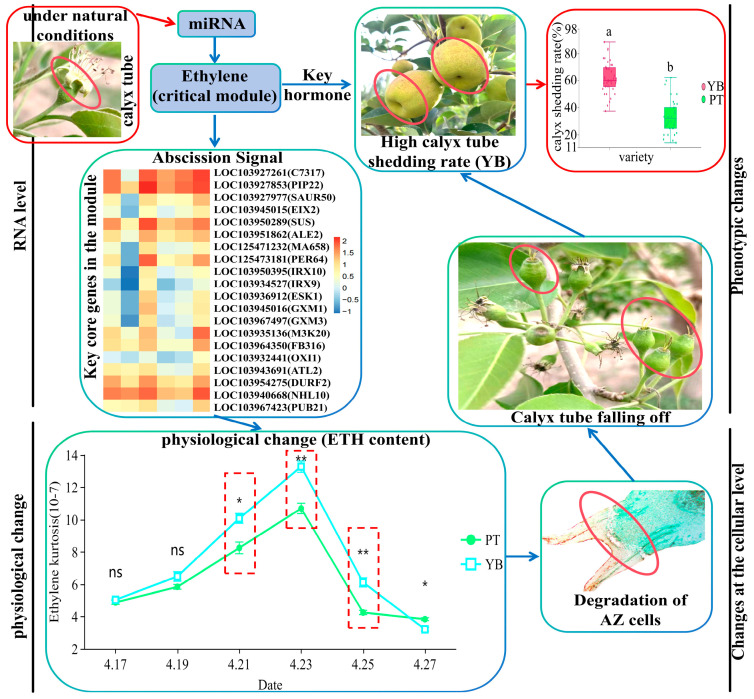
Mechanism diagram of calyx tube shedding based on WGCNA analysis. In the expression heat map of 20 key core genes, the closer the color is to red, the higher the expression level is. In the ethylene content map, the asterisk and double asterisk indicate significant differences at the *p* < 0.05 and *p* < 0.01 levels, respectively. In the statistical diagram of the calyx removal rate, a and b indicate that there is a significant difference between the two varieties. The red dotted line frame indicates the three critical periods of calyx tube shedding. ns indicates that there is no significant difference between the two varieties.

**Figure 16 plants-13-03504-f016:**
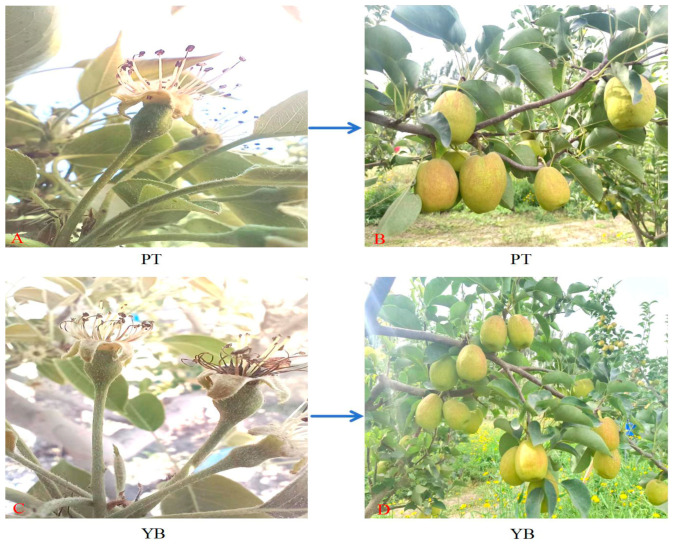
The photos of Korla fragrant pear tree and ‘Xinnonglinxiang’ tree. PT represents the Korla fragrant pear tree. YB represents the ‘Xinnonglinxiang’ tree. (**A**) is the young fruit of Korla fragrant pear tree, and (**B**) is the mature fruit of Korla fragrant pear tree. (**C**) is the young fruit of the ‘Xinnonglinxiang’ tree, and (**D**) is the mature fruit of the ‘Xinnonglinxiang’ tree.

**Figure 17 plants-13-03504-f017:**
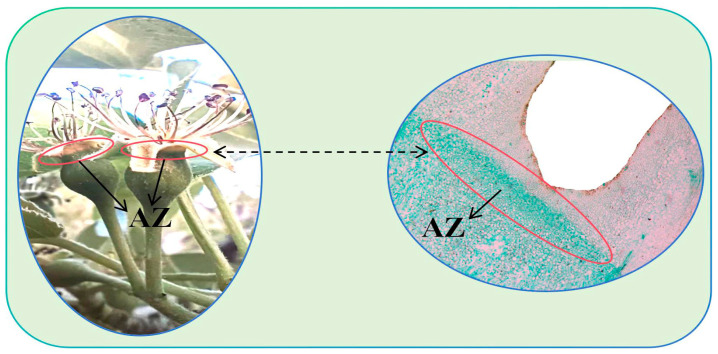
Schematic diagram of calyx tube abscission zone sampling. The **left** image is the appearance of the detachment zone, and the **right** image is the cell image of the detachment zone under the 5× objective lens.

## Data Availability

The data have been provided in the article or Appendix A.

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
