# Peer review of "Comprehensive Physiology, Cytology, and Transcriptomics Studies Reveal the Regulatory Mechanisms Behind the High Calyx Abscission Rate in the Bud Variety of Korla Pear (Pyrus sinkiangensis ‘Xinnonglinxiang’)"

_plants, 2024, doi:10.3390/plants13243504_

Round 1

Reviewer 1 Report

Comments and Suggestions for Authors

I checked your manuscript and described comments below.

Korla fragrant pear (Pyrus sinkiangensis Yu) is a pear native to the Xinjiang Uyghur Autonomous Region (China). This paper uses bioinformatics techniques to provide a very good analysis of the control mechanism of the calyx-abscission rate of this pear.

The results of this paper are at the accept level, but I think it would be better to consider the following points.

1.       Many people around the world don't know about the Korla fragrant pear (Pyrus sinkiangensis Yu). I think it would be better to include a phylogenetic tree explaining what kind of pear it is and a photo of the fruit.

2.       The analysis procedure is complicated and difficult to understand, so I think it would be better to include a flow chart.

3.       I think your abstract is too long. I think you should narrow down your main points a bit.

I don't think this paper has major problems and grammatical problems.

Author Response

Dear reviewer

Re: Manuscript ID: plants-3299642

We sincerely thank you for your suggestions on the manuscript. We would also like to thank you for your time and effort in reviewing this manuscript.

Point-to-Point Response to Reviewers’ Comments

Point 1: Many people around the world don't know about the Korla fragrant pear (Pyrus sinkiangensis Yu). I think it would be better to include a phylogenetic tree explaining what kind of pear it is and a photo of the fruit.

Response 1: We appreciate your suggestion very much. We admit that many people in the world do not understand Korla fragrant pear, because Korla fragrant pear is a native pear variety in Korla, Xinjiang, China. I am sorry for our negligence. In fact, in recent years, some researchers have studied the origin and classification of Korla pear. However, there has been no unified conclusion on the origin of Korla pear. This issue has not yet been explored. This will also be the future research direction of our team. Therefore, we do not intend to draw phylogenetic trees to illustrate the classification of Korla Fragrant Pear. This is not the focus of our study of this manuscript. In the introduction section, we give an overview of Korla pear. In addition, we added photos of Korla fragrant pear fruit and Xinnonglinxiang fruit in the experimental material 4.1 chapter. We hope that through the display of fruit photos, so that readers have a whole understanding of Korla pear. Regarding the classification and origin of Korla fragrant pears, I think our team will be committed to studying this issue in the future. Thank you again for pointing out this issue. Changes have been marked in red and their position in the revised manuscript - page number: 20; line number: 669-677.

Point 2: The analysis procedure is complicated and difficult to understand, so I think it would be better to include a flow chart.

Response 2: Thank you for pointing out this problem. We accept your suggestion. We acknowledge that the analysis process of this study is complicated. We combined the contents of physiology, cytology and transcriptomics to study the regulation mechanism of high calyx removal rate of Korla fragrant pear bud mutation variety 'Xinnonglinxiang'. In order to enable readers to understand the overall content of this study more clearly, we have made a beautiful flow chart. We hope that this flow chart can make readers realize the logical framework of this study, so as to enhance readers ' interest in reading. We put the flow chart in the supplementary material. Supplementary Figure S4 shows the logical framework and flow chart of this research. Changes have been marked in red and their position in the revised manuscript - page number: 20; line number: 679-680.

Point 3: I think your abstract is too long. I think you should narrow down your main points a bit.

Response 3: Thank you for pointing this out. We accept your suggestion. We have modified the content of the abstract. The focus of this study is to find the key genes that regulate the abscission of fruit calyx. Therefore, in the abstract part, we mainly reflect our important findings in transcriptomics. Through molecular methods, it provides reliable data for screening candidate genes for calyx shedding and analyzing the regulation mechanism of high calyx shedding rate. Finally, it provides a theoretical basis for improving fruit calyx removal rate through genetic improvement in production. Changes have been marked in red and their position in the revised manuscript - page number: 1; line number: 14-36.

We thank you again for your suggestions for improvement of our manuscripts. If there are any problems that need to be modified, please let us know. We will do our best to revise the manuscript.

Reviewer 2 Report

Comments and Suggestions for Authors

The manuscript „Comprehensive studies in physiology, cytology, and transcriptomics have revealed the regulatory mechanisms behind the high calyx-abscission rate in bud variety of Korla pear (Pyrus sinkiangensis 'Xinnonglinxiang')” concerns physiological, cytological and transcriptomic research on the pear.

The title is relevant to the content. However, in my opinion the manuscript is too long and I recommend that it be drastically shortened. Especially Results and Discussion should be shortened. These chapters should be written more concisely. The figures are carefully designed and described, but the authors should consider whether all of tchem are necessary. For example Figure 4a and Figure 4b show the same data but are presented in different ways. The same comment applies to Fig. 7, 8, 9, 12, 13, 14; graphs a,b,c show the same data. Therefore, I recommend that the figures be rearranged so that the relationships between the genes are presented in a concise manner.

The abbreviations are used in  the Abstract, Introduction and Results, e.g. ZR, ABA, CEL, WGCNA etc. They are explained in the Materials and Methods which is the chapter at the end of the manuscript.

For Xinnonglinxiang the abbreviation PT or CK is used, why?

123-124 Please give the full name of the genes or at least what kind of genes they are, what they encode.

 In Materials and methods the variety 'Xinnonglinxiang’ is very laconically described as „Korla fragrant pear bud mutation”. Which mutation? This may be important for correct interpretation especially of genetic analyses.

 Please complete the real-time analysis information. There is no information on how the RNA was isolated. Was RNA purified and reverse transcribed? Which reaction mix was used for rt PCR was used? What thermocycler model was used?

Author Response

Dear reviewer

Re: Manuscript ID: plants-3299642

We sincerely appreciate your comments and suggestions on our manuscripts. We would also like to thank you again for the time and effort you have put into the manuscript.

Point-to-Point Response to Reviewers’ Comments

Point 1: the manuscript is too long and I recommend that it be drastically shortened. Especially Results and Discussion should be shortened. These chapters should be written more concisely. The figures are carefully designed and described, but the authors should consider whether all of tchem are necessary. For example Figure 4a and Figure 4b show the same data but are presented in different ways. The same comment applies to Fig. 7, 8, 9, 12, 13, 14; graphs a,b,c show the same data. Therefore, I recommend that the figures be rearranged so that the relationships between the genes are presented in a concise manner.

Response 1: Thank you for pointing out this problem. We agree with your opinion. We have checked the content of the full text and shortened the content. In addition, we check all the graphs in the full text and delete the graphs that express the same information. We also deleted some redundant explanations of the figure. At the same time, we re-arranged the graph. We hope to present the difference in gene expression in the calyx tube abscission zone of the two varieties to the reader through a simple and beautiful map. In addition, in the discussion section, first of all, we deleted the original discussion section 3.1, which seems not to be the focus of this article. Secondly, we put the original results of chapter 2.4.7 and chapter 2.4.9 into the discussion section. These two chapters are more in line with the content of the discussion section. The focus of this study is to reveal the regulation mechanism of high calyx removal rate of Korla fragrant pear bud mutation variety ' Xinnonglinxiang ' by combining physiology, cytology and transcriptomics. We believe that the focus of this study is to discuss and summarize the key genes we found and the mechanism map we summarized. Therefore, we put chapter 2.4.7 and chapter 2.4.9 into the discussion section. In addition, we have deleted sections 3.2 and 3.3 of the original discussion. We think these two chapters are tedious and redundant. Through modification, we have greatly shortened the content of the discussion section. Thus, in the discussion section, we highlight the possible calyx shedding regulation mechanism diagram we summarized. Thank you again for pointing out this problem. Changes have been marked in red and their position in the revised manuscript - page number: 6-7; line number: 246-261; page number: 9-10; line number:324-348; page number: 11; line number: 363-376; page number: 11-12; line number: 377-394; page number: 15; line number:468-480; page number: 16-19; line number:517-647.

Point 2: The abbreviations are used in the Abstract, Introduction and Results, e.g. ZR, ABA, CEL, WGCNA etc. They are explained in the Materials and Methods which is the chapter at the end of the manuscript. For Xinnonglinxiang the abbreviation PT or CK is used, why?

Response 2: Thank you for pointing out this. In this study, in order to facilitate readers to understand, we use YB to represent 'Xinnonglinxiang', and PT to represent Korla fragrant pear. YB and PT are not abbreviations. We only use these letters to represent the corresponding variety, which is convenient for readers to distinguish. Thank you again for asking this question.

Point 3: 123-124 Please give the full name of the genes or at least what kind of genes they are, what they encode. 

Response 3: Thank you for pointing out this problem. I am sorry for our negligence. We accept your suggestion. We have supplemented the full names of these genes. Changes have been marked in red and their position in the revised manuscript - page number: 3; line number: 117-121.

Point 4: In Materials and methods the variety 'Xinnonglinxiang’ is very laconically described as Korla fragrant pear bud mutation”. Which mutation? This may be important for correct interpretation especially of genetic analyses.

Response 4: Thank you for pointing out this problem. I am sorry for our negligence. We accept your suggestion. We supplemented the introduction of bud mutation variety in the experimental material 4.1 chapter. And we cited an article that our team had previously published. And this article gives a good description of the bud mutation variety. Changes have been marked in red and their position in the revised manuscript - page number: 19-20; line number: 657-677.

Point 5: Please complete the real-time analysis information. There is no information on how the RNA was isolated. Was RNA purified and reverse transcribed? Which reaction mix was used for rt PCR was used? What thermocycler model was used?.

Response 5: Thank you for pointing out this problem. I am sorry for our negligence. We accept your suggestion. We have supplemented the relevant information. First of all, in the chapter of experimental method 4.2.4, we supplemented the related contents of RNA extraction and library construction. Then, we supplemented the specific steps of fluorescence quantitative PCR in the chapter of experimental method 4.2.7. We thank you again for pointing out this issue. Changes have been marked in red and their position in the revised manuscript - page number: 22-23; line number: 779-823; page number: 24; line number: 870-884.

If there is still a problem with the modification, please notify us. We will work harder to modify. We once again sincerely thank you for your suggestions on the manuscript.

Reviewer 3 Report

Comments and Suggestions for Authors

Dear Authors

It was with great pleasure that I read the manuscript of Xian'an Yang et al “Comprehensive studies in physiology, cytology, and transcriptomics have revealed the regulatory mechanisms behind the high calyx-abscission rate in bud variety of Korla pear (Pyrus sinkiangensis ‘Xinnonglinxiang’)”. The topic of the research is very relevant and corresponds to the profile of the Plants. New experimental results were obtained in the course of the study. A huge amount of work has been done in which the authors have made extensive use of physiological, cytological and molecular criteria and analytical techniques.

This study was aimed at identifying the regulatory mechanisms behind the high calyx-abscission rate in bud variety of Korla pear (Pyrus sinkiangensis 'Xinnonglinxiang'). Korla Fragrant Pear (YB) and 'Xinnonglinxiang' (PT) with different degrees of calyx abscission were used as model objects. Moreover, 'Xinnonglinxiang' is a mutant line of Korla fragrant pear, which has the characteristics of high calyx removal rate under natural conditions. The mechanism of high calyx removal rate of 'Xinnonglinxiang' is unclear. It is very important that physiological, structural and molecular changes during the formation of the calyx layer were studied in dynamics. I liked this manuscript, although there are some comments.

I would like to dwell on the study of the hormonal status of the detachment layer. The authors studied the content of a number of hormones (ZR, ABA, IAA, Eth and GA3) and reasonably consider them to be regulators of the calyx abscission process. In addition, hormone receptor and signaling genes were studied. An important advantage of this study was the establishment of correlation between the expression levels of “hormonal” genes and hormone content in tissues.

The assessment of hormone content raises several questions.

1.         The article absolutely does not describe the methods of determining the content of endogenous hormones, which is a drawback and should be eliminated.

2.         Why did the authors used HPLC method to analyze hormones, although in recent years journals usually publish results obtained exclusively by LC/MSn?

As far as I know, a number of leading laboratories for the analysis of hormonal status have tried to analyze cytokinins (and other hormones) by HPLC.  It was not possible to get reliable results, as the amounts of other compounds in the samples are much much higher. The only hormone which could analyzed by HPLC is ABA during drought stress. But definitely not gibberellins, cytokinins or auxin. Unfortunately for hormone analysis it is necessary to use LC/MSn or GC/MSn. I am sorry but the described method cannot give any relevant results (even if authors know the retention times of standards).

3.         When determining hormone content, it is absolutely necessary to use internal standards for each analyzed compound. This is due to the fact that in each purification step the authors had to have losses, specific for each type of hormone The question arises, what internal standards were used in this work? If no internal standards were used, then the results are not reliable.

4. The authors studied ZR. Are we talking about tZR or cZR? It also remains unclear why the authors determined the content of zeatinriboside, although there are other forms of active and inactive natural cytokinins in plants?

The manuscript, I have no doubt, can be published, but the question of the correctness of the method of determining the content of phytohormones needs to be resolved.

Kind regards

Author Response

Dear reviewer

Re: Manuscript ID: plants-3299642

We thank you for your suggestions and comments on our manuscripts. At the same time, we thank you for the energy you spent in reviewing this manuscript.

Point-to-Point Response to Reviewers’ Comments

Point 1: The article absolutely does not describe the methods of determining the content of endogenous hormones, which is a drawback and should be eliminated.

Response 1: Thank you for pointing out this problem. I am sorry for our negligence. We appreciate your suggestion very much. We have supplemented the determination method of endogenous hormone content in chapter 4.2.3 of the material method. We thank you again for pointing out this issue. Changes have been marked in red and their position in the revised manuscript - page number: 21-22; line number: 720-768.

Point 2: Why did the authors used HPLC method to analyze hormones, although in recent years journals usually publish results obtained exclusively by LC/MSn?

As far as I know, a number of leading laboratories for the analysis of hormonal status have tried to analyze cytokinins (and other hormones) by HPLC. It was not possible to get reliable results, as the amounts of other compounds in the samples are much much higher. The only hormone which could analyzed by HPLC is ABA during drought stress. But definitely not gibberellins, cytokinins or auxin. Unfortunately for hormone analysis it is necessary to use LC/MSn or GC/MSn. I am sorry but the described method cannot give any relevant results (even if authors know the retention times of standards).

Response 2: Thank you for pointing out this problem. I am sorry for our negligence. In this study, we used a combination method to determine the content of plant hormones. That is, high performance liquid chromatography-mass spectrometry. But due to our negligence, we write wrongly. We have miswritten the abbreviation of this method. The correct abbreviation is HPLC-MS. Again, I am sorry for our wrong writing. We have made corrections. Changes have been marked in red and their position in the revised manuscript - page number: 21-22; line number: 720-768.

Point 3: When determining hormone content, it is absolutely necessary to use internal standards for each analyzed compound. This is due to the fact that in each purification step the authors had to have losses, specific for each type of hormone The question arises, what internal standards were used in this work? If no internal standards were used, then the results are not reliable.

Response 3: Thank you for pointing this out. We accept your suggestion. In the determination of plant hormones, we used internal standards. These internal standards are reflected in section 4.2.3 of the Material Method. Changes have been marked in red and their position in the revised manuscript - page number: 21-22; line number: 720-768.

Point 4: The authors studied ZR. Are we talking about tZR or cZR? It also remains unclear why the authors determined the content of zeatinriboside, although there are other forms of active and inactive natural cytokinins in plants?

Response 4: Thank you for pointing this out. I am sorry for our wrong writing. In this manuscript, the type of zeatin is tZR. We have labeled this indicator in the material method. At present, there are few studies on the shedding of plant organs by zeatin, which is only found in the process of calyx shedding of 'Korla Fragrant Pear'. For example, studies have found that high levels of zeatin in the lower part of the young fruit inhibit the shedding of the calyx tube. Therefore, in order to explore the change trend of zeatin content in the process of calyx shedding of 'Xinnonglinxiang' variety, we chose to determine this index. We also discussed the reason why we chose to determine the zeatin index in the discussion section 3.1. Changes have been marked in red and their position in the revised manuscript - page number: 21; line number: 724; page number: 16; line number: 532-539.

We thank you again for your suggestions for improvement of our manuscripts. If there are any problems that need to be modified, please let us know. We will do our best to revise the manuscript.

Round 2

Reviewer 2 Report

Comments and Suggestions for Authors

The manuscript has been shortened and corrected according to my comments.

Reviewer 3 Report

Comments and Suggestions for Authors

Dear Authours,

The authors have seriously revised the manuscript. Although I still have some questions for the author, I recommend accepting the manuscript for publication in its current form, since the data obtained are interesting and fundamental.

Kind  regards